# Age-related differences in immune dynamics during SARS-CoV-2 infection in rhesus macaques

Emily Speranza[1,*], Jyothi N Purushotham[2,3,*], Julia R Port[2], Benjamin Schwarz[4], Meaghan Flagg[2], Brandi N Williamson[2], Friederike Feldmann[5], Manmeet Singh[2], Lizzette Pérez-Pérez[2], Gail L Sturdevant[2], Lydia M Roberts[4], Aaron Carmody[6], Jonathan E Schulz[2], Neeltje van Doremalen[2], Atsushi Okumura[2], Jamie Lovaglio[5], Patrick W Hanley[5], Carl Shaia[5], Ronald N Germain[1], Sonja M Best[2], Vincent J Munster[2], Catharine M Bosio[4], Emmie de Wit[2]

**Advanced age is a key predictor of severe COVID-19. To gain insight into this relationship, we used the rhesus macaque model of SARS-CoV-2 infection. Eight older and eight younger macaques were inoculated with SARS-CoV-2. Animals were evaluated using viral RNA quantification, clinical observations, thoracic radiographs, single-cell transcriptomics, multiparameter flow cytometry, multiplex immunohistochemistry, cytokine detection, and lipidomics analysis at predefined time points in various tissues. Differences in clinical signs, pulmonary infiltrates, and virus replication were limited. Transcriptional signatures of inflammation-associated genes in bronchoalveolar lavage fluid at 3 dpi revealed efficient mounting of innate immune defenses in both cohorts. However, age-specific divergence of immune responses emerged during the post-acute phase. Older animals exhibited sustained local inflammatory innate responses, whereas local effector T-cell responses were induced earlier in the younger animals. Circulating lipid mediator and cytokine levels highlighted increased repair-associated signals in the younger animals, and persistent pro-inflammatory responses in the older animals. In summary, despite similar disease outcomes, multi-omics profiling suggests that age may delay or impair antiviral cellular immune responses and delay efficient return to immune homeostasis.**

## Introduction

Increased rates of severe and fatal coronavirus disease 2019 (COVID-19) have been reported in individuals >65 yr of age, males, and those with comorbidities, including hypertension, type II diabetes, cardiovascular disease, obesity, lung disease, and renal disease ([1], [2]). Aging results in numerous changes to cells and mediators of the immune system, which alter susceptibility to infection, disease progression, and clinical outcomes. Defining features of this process, termed immunosenescence, include cytokine dysregulation, an accumulation of senescent cells leading to chronic inflammation, a loss of naïve T- and B cells, and defective responses by innate immune subsets ([3], [4]). A contribution of age-associated changes in the immune landscape to increased disease and tissue damage has been described for a variety of viral respiratory pathogens, including influenza A virus and respiratory syncytial virus ([5], [6], [7]). These studies implicate faulty or poorly regulated interactions between the immune system and the local cellular environment (e.g., respiratory epithelium) in the breakdown of protective responses to infectious agents. Ultimately, the confluence of these events may result in greater accrual of tissue damage, sustained local inflammation, severe clinical disease, and suboptimal induction of immune effector and memory responses.

In patients, the effects of age on the immune system have been studied, but these investigations have had to rely solely on profiling of circulating responses, whereas the complications of COVID-19 mainly occur in the lower respiratory tract. Analyses performed within the respiratory tract are typically restricted to samples collected post mortem. Also, it is rare to have access to pre-infection samples in patients and to subsequently follow these same individuals through the course of disease. Finally, the impact of age on clinical versus subclinical aspects of severe acute respiratory syndrome-coronavirus-2 (SARS-CoV-2) infection is an area in need of additional investigation. Animal models are an important tool for understanding the immunopathogenesis of SARS-CoV-2 because they enable concurrent assessment of immune responses at the primary site of infection, the lungs, and in

[1]Laboratory of Immune System Biology, National Institute of Allergy and Infectious Disease, National Institutes of Health, Bethesda, MD, USA   [2]Laboratory of Virology, National Institute of Allergy and Infectious Disease, National Institutes of Health, Hamilton, MT, USA   [3]The Jenner Institute, Nuffield Department of Clinical Medicine, University of Oxford, Oxford, UK   [4]Laboratory of Bacteriology, National Institute of Allergy and Infectious Disease, National Institutes of Health, Hamilton, MT, USA   [5]Rocky Mountain Veterinary Branch, National Institute of Allergy and Infectious Disease, National Institutes of Health, Hamilton, MT, USA   [6]Research Technologies Branch, National Institute of Allergy and Infectious Disease, National Institutes of Health, Hamilton, MT, USA

Correspondence: emmie.dewit@nih.gov
*Emily Speranza and Jyothi N Purushotham contributed equally to this work.

circulation through multiple phases of disease progression and recovery. The effects of age on disease outcome after SARS-CoV-2 infection have been explored in nonhuman primate species ([8], [9], [10], [11], [12]). Consistent with findings in humans, these studies suggest that advanced age may also be associated with poorer outcomes to SARS-CoV-2 infection in these models. However, these studies have largely not addressed the relative roles of dysregulated immune responses versus virus-induced damage in driving differences in outcomes related to age. Furthermore, a potential link between underlying, age-associated immunological changes and subclinical effects of infection has not been described in detail.

To further characterize the relationship between age and host immune responses to SARS-CoV-2 infection, we conducted a time-resolved evaluation of disease in age-stratified cohorts of rhesus macaques. We performed extensive local and systemic sampling throughout the course of disease through swabbing and collection of blood, tissue, and bronchoalveolar lavage fluid (BALF). Doing so facilitated the detection of immune-related differences, using single-cell and bulk approaches, at the site of infection during critical stages of disease progression: the acute phase, the post-acute phase, and the transition point between the two. Age-related divergences in inflammation, immune regulation, and adaptive immunity appeared several days after inoculation and were exacerbated over time. Our study represents a unique, in-depth kinetic evaluation of the interaction between infection and the aging immune system in a nonhuman primate model.

# Results

## Multi-omics assessment and baseline stratification of rhesus macaque cohorts

We used a multi-omics approach to profile local and systemic immune responses to SARS-CoV-2 infection in aged (mean age = 18 yr) and sub-adult (mean age = 3.4 yr) rhesus macaques. Animals were inoculated with $2.6 \times 10^6$ TCID$_{50}$ of SARS-CoV-2 and clinical, virological, and immunological parameters were monitored over time using multiple modes of assessment on various sample types ([Fig 1A]). Four animals from each age-group were euthanized at 7 and 21 days post inoculation (dpi) to perform tissue-specific analyses during the acute and post-acute stages of infection.

The older animals were classified as overweight according to body weight and body condition scores (BCS) ([Fig 1B]). We characterized the circulating lipid content at baseline because lipid mediators (LMs) are important immune regulators that have been shown to influence COVID-19 severity in humans ([13]). Liquid chromatography tandem mass spectrometry (LC-MS/MS) was used to target ~1,200 individual lipid species across glycerolipids, cholesterol-esters, sphingolipids, phospholipids, and free fatty acids. Upon comparison of the summation of signals from each lipid class, we noted that the older animals displayed elevated levels of neutral ($P = 0.03$), phospho- ($P = 0.03$), and lyso-phospho- ($P = 0.03$) lipid classes ([Fig 1C]). However, we did not detect the disruption of sphingolipid species ($P = 1$), which were a primary marker of circulating lipid disruption in obese nonhuman primates ([14]) suggesting the older animals were overweight but not obese.

Altered immune states in older individuals may include the development of a chronic inflammatory phenotype (also known as

"inflammaging") characterized by a persistent elevation of pro-inflammatory cytokine levels ([15]). Consistent with this phenotype, concentrations of serum IL-6 ($P = 0.03$) were increased in older versus younger animals at pre-infection ([Fig 1D]). However, concentrations of other cytokines associated with chronic inflammation, like TNF-$\alpha$ ($P = 0.70$) and IL-1RA ($P = 0.88$), were not increased ([Fig S1]) ([16], [17]). Although not as strongly differentiated, IL-5 ($P = 0.12$) and IL-8 ($P = 0.07$) were detected at lower concentrations in the older than in younger animals ([Fig 1D]), which may reflect a reduced capacity for regulating innate cell chemotaxis ([18]).

Declining adaptive immunity is another hallmark of immuno-senescence. A reduction in thymic and bone marrow function contributes to a loss of naïve T- and B cells and the accumulation of terminally differentiated effector cells ([19], [20], [21], [22]). Consistent with other aging models, the older rhesus macaques exhibited lower frequencies of circulating naïve (CD28$^+$CD95$^{lo}$) CD4$^+$ ($P = 0.0006$) and CD8$^+$ ($P = 0.0002$) T cells, as well as naïve (IgD$^+$ CD27$^-$ CD21$^+$) ($P = 0.005$) and total CD20$^+$ ($P = 0.01$) B cells. Corresponding elevated frequencies of effector/effector memory (CD28$^-$ CD95$^+$) CD4$^+$ ($P = 0.01$) and CD8$^+$ ($P = 0.001$) T cells, along with plasma cells (CD20$^-$ CD38$^+$ CD138$^+$) ($P = 0.005$) were also observed in the older animals ([Fig 1E and F]). The collective results of our baseline sampling revealed age-related differences in body condition, lipid metabolism, cytokine regulation, and adaptive immunity with the potential to impact responses to infection.

## Moderate age-related differences in clinical and virological outcomes after SARS-CoV-2 inoculation

To assess differences in the progression of SARS-CoV-2 infection, we conducted standardized clinical scoring, scoring of pulmonary infiltrates on radiographs, and virological measurements in respiratory tract samples. Older animals displayed slightly elevated clinical scores over the course of infection (area under the curve [AUC] = 150.2 ± 16.86 in older animals versus AUC = 75.25 ± 9.45 in younger animals; $P < 0.0001$) ([Fig 2A]). By 21 dpi, three of four younger animals had fully recovered, whereas mild clinical signs could still be observed in all older animals (Table S1). Increased pulmonary infiltrates were observed on radiographs in the older than in the younger animals (AUC = 19.88 ± 6.3 in older animals versus 7 ± 2.53 in younger animals; $P = 0.06$) ([Fig 2B]). Pulmonary infiltrates were no longer detected in either cohort by 10 dpi, suggesting that respiratory disease was limited to between 1 and 7 dpi ([Fig 2B]).

Virus replication in the upper and lower respiratory tract was assessed using nose swabs, throat swabs, BALF, and lung tissue. Viral loads were quantified using qRT-PCR for the detection of genomic RNA (gRNA) and subgenomic RNA (sgRNA) as a measure of recent virus replication. gRNA and sgRNA clearance in nose and throat swabs over the time course did not differ between groups ([Fig 2C and D]). In the lower respiratory tract, viral gRNA clearance from BALF was relatively consistent between the younger animals (mean of change = −1.6, SD of change = 0.3), whereas substantial variability was observed within the older cohort, as evidenced by high variance in the mean change in viral load between 3 and 7 dpi (mean of change = −1.3, SD of change = 1.13) ([Fig 2E]). Meanwhile, detection of sgRNA in BALF revealed similar control of virus replication, regardless of age ([Fig 2E]). Likewise, although comparison of gRNA in lung tissue at 7 dpi suggested that the younger animals

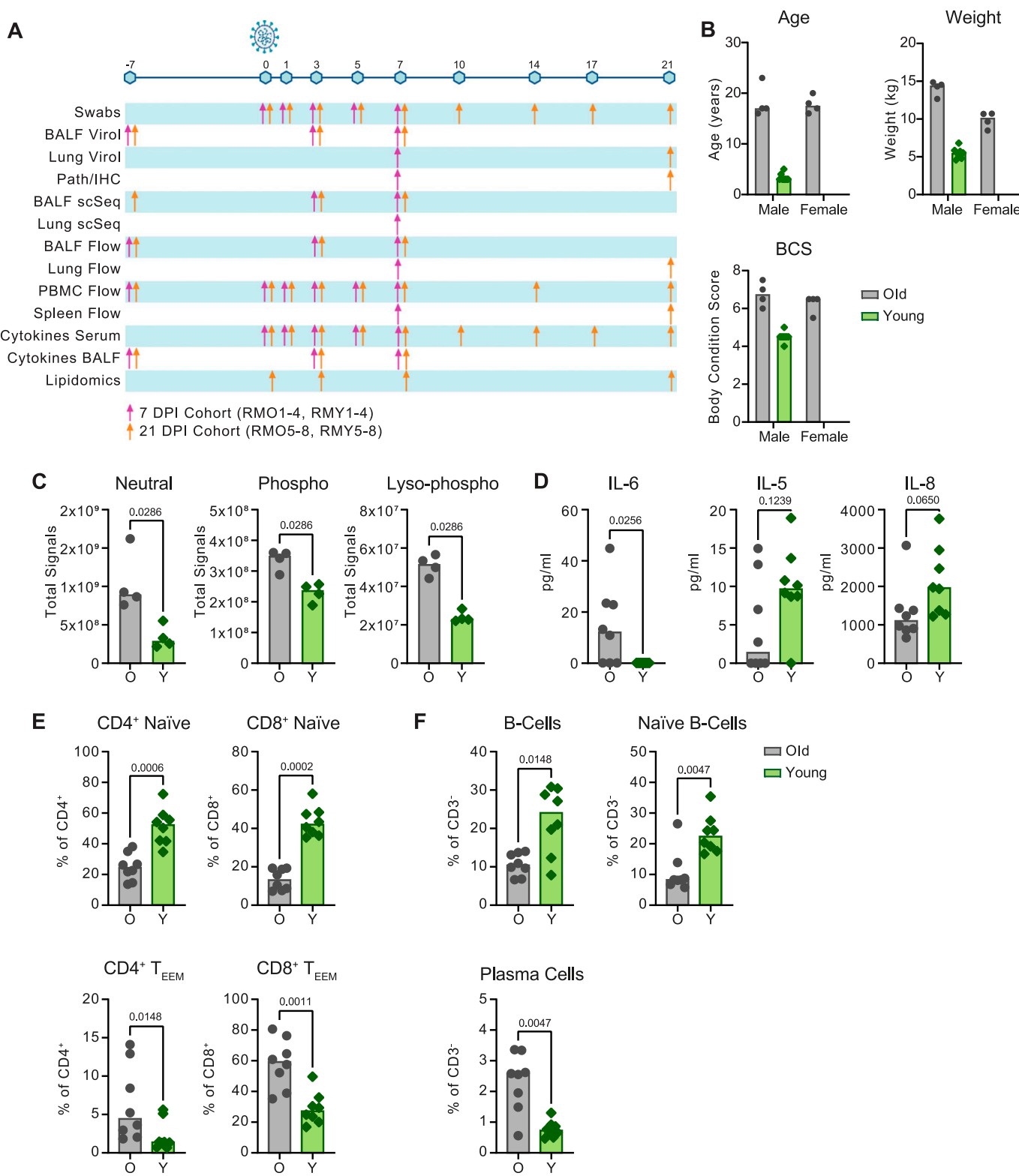

**Figure 1. Multi-omics data collection and baseline comparison between old and young rhesus macaques.**
**(A)** Overview of samples collected. Sampling time points are depicted across the timeline as days post inoculation (dpi). The colored arrows indicate sampling performed on animals necropsied at 7 dpi (pink) or animals necropsied at 21 dpi (orange). BALF = bronchoalveolar lavage fluid, IHC = immunohistochemistry. **(B)** Summary of age, weight and body condition score divided by sex. **(C, D, E, F)** Baselines of total lipid signal by lipid class (C), serum cytokines (D), circulating T-cell subsets (naïve and effector/effector-memory [T$_{EEM}$]) (E), and circulating B-cell subsets (F) present at different concentrations (C, D) or frequencies (E, F) pre-inoculation (−7 or 0 dpi). Grey represents older and green represents younger rhesus macaques. *P*-values were calculated using the Mann–Whiney *U* test.

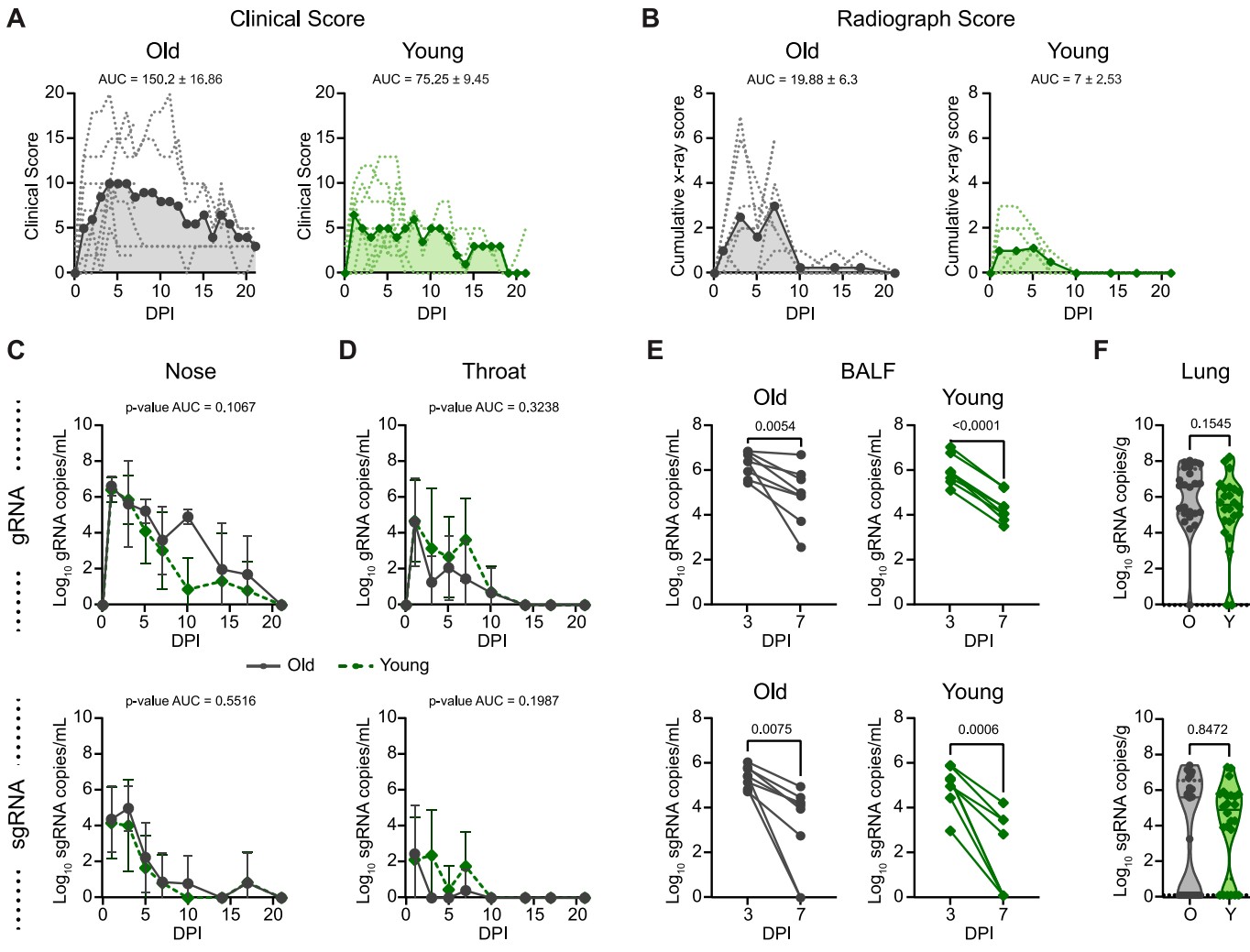

**Figure 2. Comparison of clinical and virological outcomes after SARS-CoV-2 inoculation.**
**(A)** Comparison of clinical scores (y-axis) over the time course of infection (x-axis). Data from individual animals are indicated by dashed lines, group means are shown using solid lines, and the area under the curve is represented by shading. **(B)** Ventro-dorsal and lateral radiographs were taken on clinical examination days and scored for the presence of pulmonary infiltrates by a clinical veterinarian according to a standardized scoring system. **(A)** Individual lobes were scored and scores per animal per day were totaled and displayed in the same format as panel (A). **(C)** Genomic RNA (gRNA, top) and subgenomic RNA (sgRNA, bottom) detected in nasal swabs after inoculation. Data points indicate the geometric mean; error bars represent SD. **(D)** gRNA and sgRNA quantification in throat swabs. **(E)** gRNA and sgRNA quantification in BALF. Matched values for individual animals at 3 and 7 days post inoculation are indicated by a connecting line. **(F)** gRNA and sgRNA quantification in lung samples collected at 7 days post inoculation. Points represent a single lung lobe section in an individual animal. Grey represents older (O) and green represents younger (Y) rhesus macaques. **(C, D, E, F)** P-values were calculated using an unpaired t test comparing the area under the curve values (C, D), paired t test (E), or Mann–Whiney U test (F).

cleared virus material (gRNA P = 0.15) slightly more efficiently, levels of sgRNA did not differ between age cohorts (sgRNA P = 0.8) (Fig 2F). Despite the lack of a difference in viral clearance, older animals showed a more rapid induction of circulating IgG responses (Fig S2) as early as 14 dpi (P-value = 0.001). However, circulating IgG titers were low or undetectable in the animals of both cohorts until after the virus was cleared from the respiratory tract (beginning at 10 dpi).

### Divergence in immune and lung parenchymal cell transcriptional responses begins at the transition to the post-acute phase of SARS-CoV-2 infection

We performed single-cell RNA sequencing in BALF (n = 4 animals per age-group at –7 dpi, n = 8 animals per age-group at 3, and 7 dpi), and

lung tissue (n = 4 animals per age-group at 7 dpi) (Fig 1A) to profile responses to SARS-CoV-2 infection in the lower respiratory tract. All major cell types were identified in BALF and lung tissue using a computational identification algorithm (23) (Figs 3A and S3A and B). Of note, we did not detect enough reads mapping to the viral genome to determine which cell types are productively infected. Changes in cell frequencies over time were calculated in BALF; increases in the frequencies of B- and T cells in BALF were observed at 3 dpi as compared to baseline (P = 0.03 for both) and were maintained to 7 dpi. The frequency of plasmacytoid dendritic cells (pDCs) in BALF peaked at 3 dpi (P < 0.0001) and returned to baseline levels by 7 dpi, suggesting a rapid resolution of type I interferon response–stimulated cell recruitment in the lungs. Macrophage/ monocyte frequencies were depressed in BALF at 3 dpi (P = 0.01) and

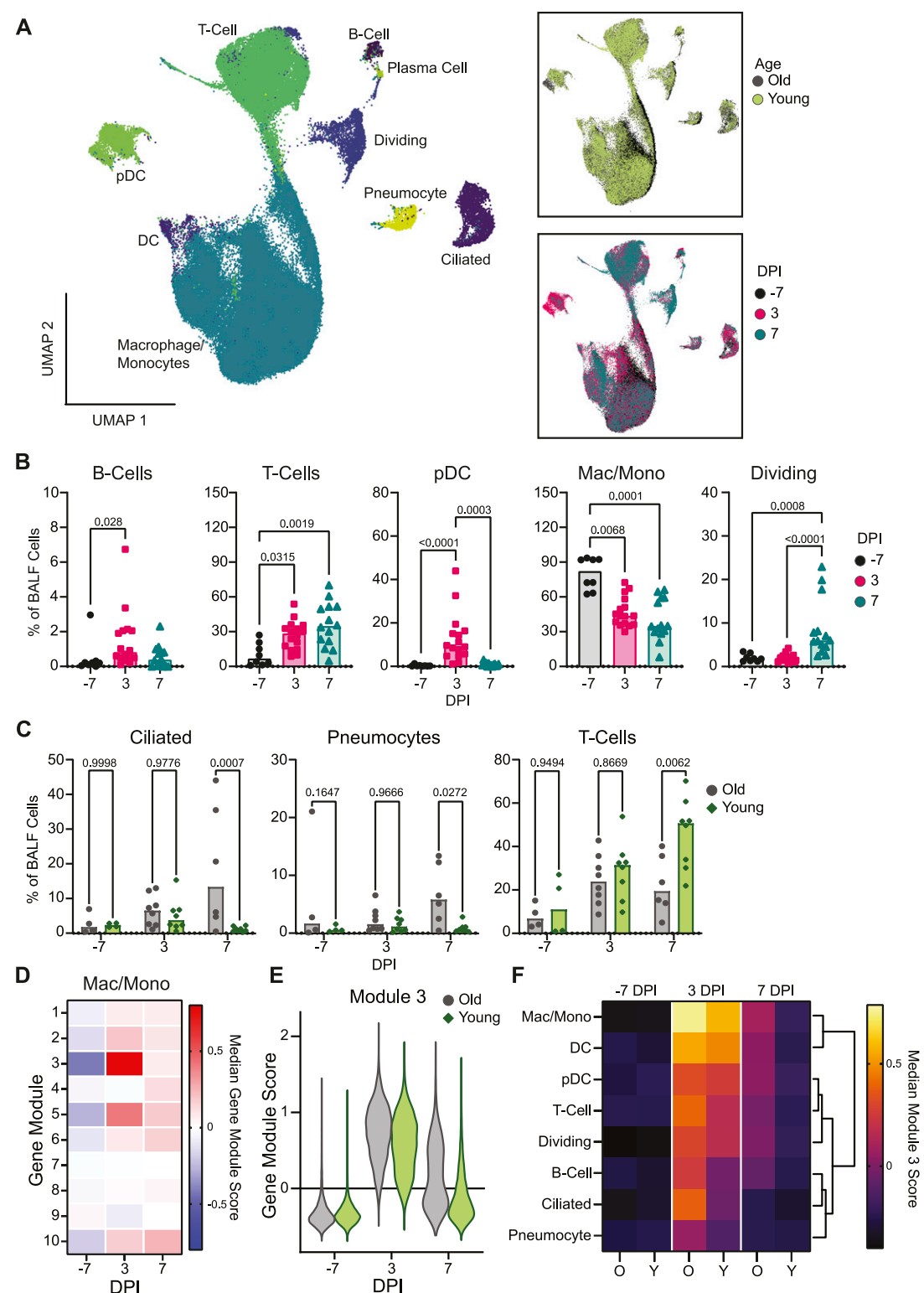

**Figure 3. Single-cell sequencing of BALF shows evidence of sustained inflammation in older rhesus macaques.**
**(A)** Uniform manifold approximation projection (UMAP) of single-cell RNA sequencing from BALF. Each point is an individual cell colored by cell type (left), age (top right), or sampling day post inoculation (dpi; bottom right). **(B)** Cell frequencies as a fraction of total BALF cells for individual animals over the course of infection for B cells, T cells, pDCs, macrophages/monocytes (mac/mono), and dividing cells. Bars represent the median for each time point. *P*-values are calculated in a two-way ANOVA. All *P*-values < 0.1 are shown. **(C)** Cell frequencies as a fraction of total BALF cells for individual animals over time separated by age-group for T cells, pneumocytes, and ciliated cells. Bars represent the mean for each group. *P*-values are calculated in a two-way ANOVA and all *P*-values are shown. **(D)** Heat map of the median gene module scaled expression value for the 10 computationally determined gene modules over the course of infection in macrophage/monocyte cells independent of age-group.

7 dpi (*P* = 0.0001), which may have been due in part to a concurrent substantial increase in the frequency of T cells (Fig 3B). A similar pattern of changes in macrophage, lymphocyte, and neutrophil frequencies was observed through flow cytometric analysis of BALF (Fig S4A). Markers of activation and/or trafficking were present on macrophages (CD86) and lymphocytes (HLA-DR, CD86, CD80, CD49d, and CCR7), with peak expression observed at 3 dpi (Fig S4B), providing further evidence of the early amplification of local signals associated with general immune cell activation and adaptive cell priming.

Age-related differences in cell frequencies were observed at 7 dpi, with the older animals displaying higher frequencies of epithelial populations, specifically ciliated cells (*P* = 0.0007) and pneumocytes (*P* = 0.03) (Fig 3C) in BALF. As expected, frequencies of pneumocytes in lung tissue did not change (*P* = 1) (Fig S3C). In addition, the frequency of BALF T cells was elevated in the younger versus older animals (*P* = 0.01), either due to enhanced expansion of local populations or systemic recruitment (Fig 3C).

To evaluate changes in the transcriptional states of specific immune cell subsets, we applied a clustering bias method to identify cell types (or sub-clusters) in BALF that showed a strong transcriptional shift after SARS-CoV-2 inoculation. We subsequently selected the two cell types demonstrating the strongest transcriptional shifts post inoculation, macrophages and dividing cells, for further assessment (Fig S3D). We then conducted gene module analysis to explore drivers of the observed transcriptional differences, starting in the macrophages. Gene expression in the macrophages/monocytes was averaged for each time point and clustered using 1,000 iterations of k-means clustering into 10 distinct gene modules (Fig 3D). Gene modules 3 and 5 showed a strong increase in expression post inoculation. Gene module 3 comprised type I interferon–responsive genes characteristic of stimulated macrophages/monocytes reacting to an active viral infection, such as *ISG15*, *IFIT2*, and *IRF7* (Supplemental Data 1). Gene module 5 was an expansion of this gene set, revealing a pro-inflammatory state within the macrophages/monocytes (e.g., *CXCL10*, *INFGR2*, and *CCL8*), which peaked at 3 dpi and began to return to baseline by 7 dpi (Fig S5 and Supplemental Data 1). This gene set was similar to ones detected through bulk RNA sequencing, yet attribution of the signature to specific cell types was not done previously (12). Individual gene module scores for the cells derived from the older versus younger animals were calculated to compare differences in median expression. Differences in expression between the older and younger animals were moderate at pre-infection (estimated *P*-value for difference in medians of 1,000 iterations = 0.02). At 3 dpi there was an elevation in gene module 3 in the older than in younger animals in the macrophages (estimated *P*-value for difference in medians out of 1,000 iterations = 0). This pattern persisted to 7 dpi (estimated *P*-value for difference in medians out of 1,000 iterations = 0) (Fig 3E). A similar age-related pattern was observed in gene module 5 (Fig S5).

To determine if the extended pro-inflammatory state in macrophages/monocytes in BALF observed in the older animals was a cell type–specific phenomenon or a consequence of general inflammation in the lower respiratory tract, we monitored the expression of gene module 3 in all cell types detected in BALF. In the older animals, there was elevated expression of module 3 at 3 dpi in all cell types, with macrophages/monocytes displaying the highest expression of this gene set, suggesting generalized lung inflammation. By 7 dpi, there was sustained expression of this gene module in all immune cells of the older animals; however, the nonimmune cells (ciliated cells and pneumocytes), had returned to baseline values (Fig 3F). In the younger animals, there was up-regulation of this gene module broadly across all immune cells; however, little up-regulation of this gene module was observed at 3 dpi in the nonimmune cells (ciliated cells and pneumocytes). In addition, all cell types had returned to near baseline levels by 7 dpi, suggesting more efficient regulation of the inflammatory response in the lungs of the younger animals (Fig 3F).

### Efficient induction of cellular immune responses in the lungs of younger animals at 7 dpi

Next, we analyzed the transcriptional changes occurring in dividing cells in BALF, the second cell type identified in the clustering bias analysis described above from the scRNA-Seq data. The dividing cell cluster was defined as containing more than 98% of cells in either the S or G2M phase of cell division. Six unique dividing cell sub-clusters were identified: CD8[+] T cells, CD4[+] T cells, macrophages with a tissue-resident phenotype, macrophages with a monocyte-like or non-tissue resident phenotype, and plasma cells (Figs 4A and S6). Cell dynamics in general showed a decrease in macrophage populations and an increase in T cells over time in the dividing cluster (Fig 4B). Differentiating the changes occurring in the dividing cell subset in the two age cohorts revealed slightly higher levels of dividing macrophages in the older animals at baseline (cluster 1, *P* = 0.04; Fig 4C). At 7 dpi, a moderate increase in nonresident-like cells was observed in the older animals (cluster 4, *P* = 0.07; Fig 4C), which is consistent with the data (Fig 3F) suggesting prolonged inflammation in the lungs of the older cohort. Meanwhile, an increase in the frequency of dividing CD8[+] T cells was selectively detected in the younger animals at this time (cluster 0, *P* = 0.01). Together, this suggests that in addition to achieving better control over inflammation, the younger animals may also demonstrate more efficient local induction or systemic recruitment of T cells than the older animals.

To determine if there was indeed a more rapid induction of T-cell responses in the younger animals as suggested by the sequencing data, we used multiparameter flow cytometry to profile T cells isolated from lung tissue at 7 dpi. Consistent with observations from single-cell sequencing, at 7 dpi, the younger animals exhibited higher frequencies of total (CD3[+]) T cells than the older animals

---

Blue colors represent low expression and pink colors represent high expression values. **(E)** Gene module score for gene module 3 for individual cells over the course of infection in the macrophages/monocytes separated by age-group. **(F)** Median gene module score for gene module 3 over the course of infection separated by age-group (O: older animals; Y: younger animals), across all cell types. The brighter the color, the higher the expression value. Columns are grouped by dpi and rows are clustered based on Euclidian distance (dendrogram right).

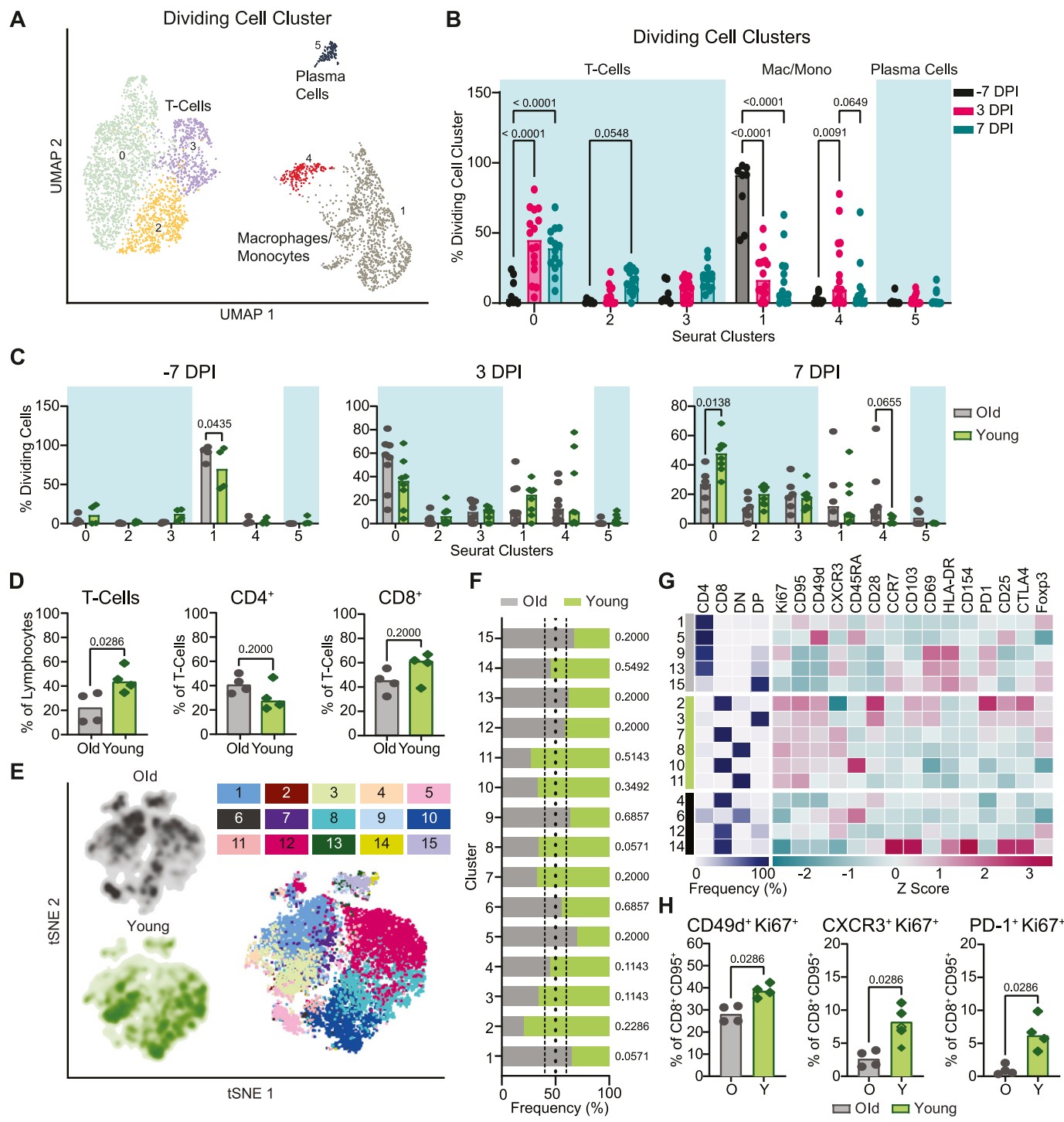

**Figure 4. Single-cell RNA sequencing and flow cytometric analyses demonstrate an age-dependent divergence in T-cell responses at 7 dpi.**
**(A)** UMAP projection of the dividing cells in BALF. Individual points represent a single cell with color determined by Seurat clusters (0–5). Labels represent the broader cell classification. Cluster 0: CD8⁺ T cells; cluster 1: macrophages with a tissue-resident phenotype; cluster 2 and 3: CD4⁺ T cells; cluster 4: macrophages with a monocyte-like or non-tissue resident phenotype; cluster 5: plasma cells. **(B)** Change in frequency of the total dividing cells for each of the Seurat clusters separated by sampling time point. Bars represent the median value. Blue boxes represent cell type groups. *P*-values are calculated in a two-way ANOVA. Only *P*-values < 0.1 are shown for clarity. **(C)** Cell frequency separated by age-group at –7 dpi (left), 3 dpi (middle), and 7 dpi (right) across the Seurat clusters as a percentage of the total dividing cell cluster. Bars represent the mean and error bars represent the SEM. *P*-values are calculated in a one-way ANOVA. Only *P*-values < 0.1 are shown. **(D)** Flow cytometric evaluation of single cell suspensions from lung tissue collected at 7 dpi. Frequencies of T cells as a percentage of lymphocytes are shown (left panel) as well as frequencies of CD4⁺ (middle panel) and CD8⁺ T cells (right panel) as proportions of total T cells. Bars depict median frequencies. *P*-values are calculated using Mann–Whiney *U* tests. **(E)** Visualization of concatenated T-cell subsets in lungs on 7 dpi across animals by t-SNE analysis. Density plots (left) show the distribution of T cells derived from each age cohort. Dot plot depicts individually color-coded FlowSOM meta-clusters overlaid onto total combined T cells (right). **(E, F)** Stacked bar chart depicting the cumulative proportion of

($P$ = 0.03), although this could not be completely attributed to differences in CD4$^+$ or CD8$^+$ T cells in either group ($P$ = 0.2 for both) (Fig 4D). We compared phenotypic or functional states within the T-cell compartment to elucidate potential age-related trends. Data from a randomly down-sampled subset of CD3$^+$ T cells was concatenated across animals and visualized by t-distributed stochastic neighbor embedding (t-SNE). The unbiased identification of 15 meta-clusters was carried out using the FlowSOM algorithm based on the inclusion of 17 markers (Fig 4E) (24). The composition of each cluster according to the frequency of cells derived from each age cohort, was defined (Fig 4F).

To differentiate phenotypes present in the older versus younger animals, we selected clusters from the flow cytometry data in which at least 60% of cells were derived from either older or younger animals and distinguished these as demonstrating cohort-specific enrichment. Application of this threshold yielded five clusters enriched in cells from the older animals, six clusters enriched in cells from the younger animals, and four clusters not notably enriched for the cells in either cohort (Fig 4F and G). The major T-cell subset classifying each cluster was delineated according to the relative frequency of CD4$^+$ (CD4$^+$ CD8$^-$), CD8$^+$ (CD4$^-$ CD8$^+$), double negative (DN, CD4$^-$ CD8$^-$), and double positive (DP, CD4$^+$ CD8$^{lo}$) T cells. Older cohort clusters were primarily CD4$^+$ T-cell populations and one DP T-cell population. Meanwhile, clusters ascribed to the younger cohort consisted of CD8$^+$, DN, and DP T cells (Fig 4G). The median fluorescent intensities (MFIs) of 15 phenotypic markers were determined within each cluster and z-scores calculated to compare relative expression across clusters. The clearest trend differentiating the clusters enriched in cells from each age cohort, was the relative increased expression of markers associated with proliferation (Ki67), antigen experience (CD95), and effector function (CD49d and to some extent CXCR3) in the clusters enriched for younger animals' cells, half of which were identified as CD8$^+$ T-cell populations (Figs 4G and S7A). These patterns may reveal expansion and differentiation occurring within the T-cell compartment of the younger animals at 7 dpi, particularly in CD8$^+$ T cells, that were not concurrently apparent in that of the older animals. Trends identified from the clustering analysis were confirmed to be representative of broader age-specific differences in T-cell phenotypes by direct comparison of the fluorescence intensities of markers on total CD8$^+$ T cells aggregated across all animals in each age cohort. The median of the fluorescence intensities of CD95, CD49d, Ki67, CXCR3, and PD-1 were greater on the CD8$^+$ T cells of younger compared with older animals ($P$ < 0.0001) (Fig S7A). We also compared frequencies of populations of CD8$^+$ T cells displaying an antigen-experienced effector phenotype and undergoing active proliferation. As a percentage of non-naïve (CD95$^+$) cells, CD49d$^+$ Ki67$^+$, CXCR3$^+$ Ki67$^+$, and PD-1$^+$ Ki67$^+$ populations were present at

higher frequencies in CD8$^+$ T-cell subsets of the younger animals in comparison with the older animals ($P$ = 0.03 for all comparisons) (Fig 4H) consistent with the results from previous analyses that were suggestive of a more robust, early CD8$^+$ T-cell response in the younger animals. Finally, to gauge whether these differences reflected the activities of bystander or virus-specific T cells, we performed an activation induced marker (AIM) assay to estimate the frequencies of SARS-CoV-2 spike protein-responsive CD4$^+$ and CD8$^+$ T cells using a combination of OX40 and CD137 surface markers. In support of our previous findings, we detected elevated frequencies of AIM$^+$ (OX40$^+$ CD137$^+$) T cells, particularly CD8$^+$ T cells ($P$ = 0.03), in the lungs of the younger animals at 7 dpi in comparison with the older animals (Fig S8).

### Pronounced age-related divergence in cellular immune responses in the lungs at 21 dpi

We next investigated the dynamic shifts in the T-cell response in the lungs in the period following virus clearance, at 21 dpi. Median T-cell frequencies in the older animals at 21 dpi appeared to increase relative to those observed at 7 dpi, potentially indicating a late expansion of local T cells or influx of T cells from peripheral blood in this cohort. Conversely, between 7 and 21 dpi, the frequency of T cells in the lungs of younger animals reduced to levels lower than those present in the older animals ($P$ = 0.03; Fig 5A). The frequency of CD4$^+$ ($P$ = 0.48) and CD8$^+$ ($P$ = 0.69) T cells within the total CD3$^+$ population was similar between cohorts (Fig 5A).

Applying the same FlowSOM clustering analysis described above to the 21 dpi datasets, we identified six clusters enriched for cells from the older animals and six clusters enriched for cells from the younger animals (Fig 5B and C). Notably, at this later time point, the segregation of cells from each age cohort into distinct clusters was more exaggerated than at 7 dpi (Fig 5C). The clusters composed of cells from the older animals spanned all major subsets: CD4$^+$, CD8$^+$, DN, and DP (Fig 5D). T cells in the younger animals were again largely found in CD8$^+$ or mixed CD8$^+$ and DN clusters (Fig 5D). Comparisons of the z-scores of phenotypic marker MFIs revealed the increased expression of Ki67, CD95, and CD49d in older animals (Figs 5D and S7B). This combination of markers matched those demonstrating increased intensity of expression on the clusters enriched for cells from the younger animals at 7 dpi, further supporting the occurrence of an age-associated delay in the acquisition of an expanded pool of antigen-experienced effectors.

Comparison of the fluorescence intensity of these markers across total CD4$^+$ and CD8$^+$ T cells from each age cohort confirmed that intensities of CD95, CD49d, and Ki67 were elevated for the CD4$^+$ T cells of the older versus younger animals ($P$ < 0.0001; Fig S7B). This effect was also maintained for CD95 and CD49d on CD8$^+$ T cells

---

cells from animals of the older and younger cohort comprising each meta-cluster from panel (E) (1–15). Vertical dotted lines represent (from left to right) the threshold of 10% for enrichment of cells from younger animals, equal composition, and threshold of 10% enrichment of cells from older animals. *P*-values comparing the median frequency of cells per animal within each cohort are listed to the right of the stacked-bars for each cluster and are calculated using Mann–Whiney *U* tests. **(G)** Heat map representation of meta-cluster phenotypes. The clusters enriched for the cells of the older (grey), younger (green), and neither (black) are indicated (left column). The major T-cell subset characterizing each cluster is shown by depicting the frequencies of CD4$^+$, CD8$^+$, double negative (DN), and double positive (DP) T cells composing each cluster (middle), as well as a phenotypic comparison of clusters by calculation of the z-score of the median fluorescence intensity of 15 markers across clusters (right). **(H)** Frequencies of expanding, effector T-cell populations as a percentage of non-naïve (CD95$^+$) and CD8$^+$ T cells. Bars depict median frequencies. *P*-values are calculated using Mann–Whiney *U* tests. In panels separated by the age-group, grey represents older (O) and green represents younger (Y) rhesus macaques.

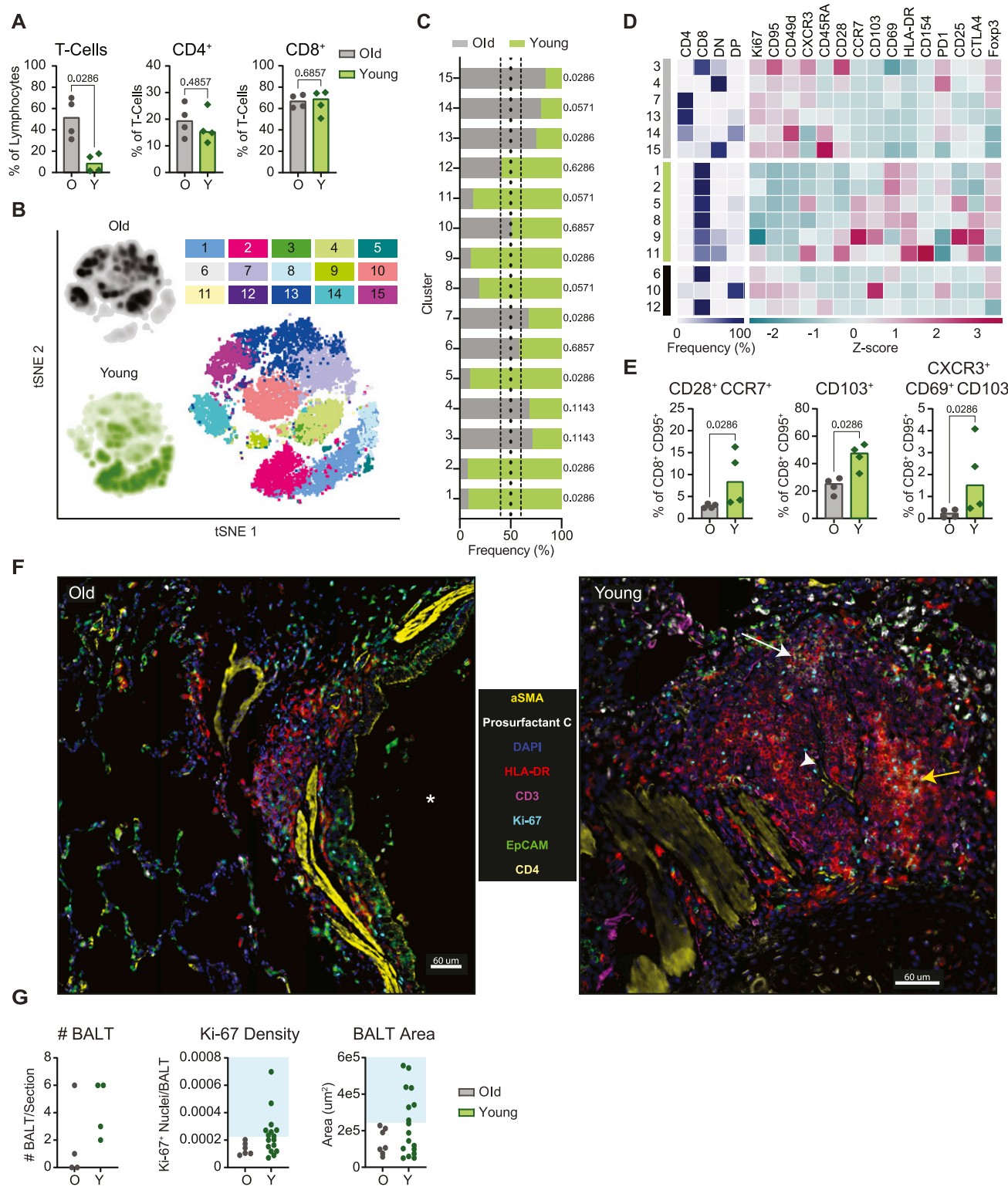

**Figure 5. Age-related differences in T-cell memory–associated phenotypes and lymphoid tissue formation are apparent in the lungs at 21 dpi.**
**(A)** Flow cytometric evaluation of single cell suspensions from lung tissue collected at 21 dpi. Frequencies of CD3+ T cells as a percentage of lymphocytes, and frequencies of CD4+ and CD8+ T cells as proportions of CD3+ T cells are shown. Bars depict median frequencies. *P*-values are calculated using Mann–Whiney *U* tests. **(B)** Data visualization by t-SNE analysis and depiction of meta-clusters generated by FlowSOM. Density plots (left) show the distribution of T cells derived from each age cohort. Dot plot depicts individually color-coded FlowSOM meta-clusters overlaid onto total combined T cells (right). **(C)** Stacked bar chart depicting the cumulative proportion of cells from animals of the older or younger cohort comprising each meta-cluster (1–15). Vertical dotted lines represent (from left to right) the threshold of 10% for enrichment of cells from younger animals, equal composition, and threshold of 10% for enrichment of cells from older animals. *P*-values comparing the median

($P$ < 0.0001), but median intensities of Ki67 on CD8$^+$ T cells, and CXCR3 and PD-1 on CD4$^+$ and CD8$^+$ T cells were higher in the younger animals ($P$ < 0.0001; Fig S7B). Meanwhile, CD8$^+$ T-cell clusters enriched for cells from the younger animals featured elevated expression of CCR7, CD103, CD69, and HLA-DR (Fig 5D). CCR7 is a marker for lymph node homing, whereas CD103 and CD69 are up-regulated on CD8$^+$ T cells with a tissue-resident memory phenotype. Cluster-specific trends were again confirmed by comparison of the fluorescence intensity of these four markers on total CD8$^+$ T cells from the younger and older animals, which verified increased expression on cells from the younger animals ($P$ < 0.0001; Fig S7C). Younger animals demonstrated elevated frequencies of non-naïve (CD95$^+$) CD8$^+$ T cells exhibiting a central memory phenotype (CD28$^+$ CCR7$^+$) in comparison with the older animals ($P$ = 0.03, Fig 5E) (25, 26, 27, 28). We additionally detected elevated frequencies of CD8$^+$ CD95$^+$ T cells in the younger cohort that were also CD103$^+$ ($P$ = 0.03) or CXCR3$^+$ CD69$^+$ CD103$^+$ ($P$ = 0.03), which may signify specific populations of effector memory cells with a tissue-resident phenotype (Fig 5E). Together, these data appear to indicate the establishment of long-term memory populations in the CD8$^+$ T-cell compartment of the younger animals at an earlier time point than in the older animals.

## Extensive bronchus-associated lymphoid tissue (BALT) formation in the lungs of young rhesus macaques

To further evaluate differences in the local cellular immune response at 21 dpi, we performed multiplex immunohistochemistry (mIHC) of lung tissue samples. We found evidence of enhanced expansion of BALT structures in the lungs of the younger animals compared with the older animals. Previous studies suggest that BALT expansion may be induced shortly after pulmonary infection (29, 30). BALT formation was primarily found around large airways (denoted by EpCAM and aSMA; Fig 5F); within the BALT we identified cells co-expressing CD3, HLA-DR, and Ki67, characteristic of activated and proliferating T cells (31). BALT structures were less abundant in the older animals, and those present were generally smaller in size and exhibited reduced proliferative activity (i.e., lower Ki67 density), whereas younger animals showed a mixture of small and larger BALT that contained germinal center-like structures with a high density of HLA-DR$^+$ CD3$^-$ Ki67$^+$ cells (Fig 5F and G). Furthermore, in contrast to those of the younger animals, the lungs of the older animals did not exhibit signs of the advanced-stage development of full lymphoid-like tissue structures, such as endothelial vessel

formation and CD3$^+$ CD4$^+$ cells surrounding clusters of CD3$^+$ HLA-DR$^+$ (representing activated T cells) and CD3$^-$ HLA-DR$^+$ (comprising antigen-presenting cells and B cells) populations (Fig 5F). However, it is possible that similarly mature BALT would have appeared in the lungs of older animals if given more time. In combination with the kinetic differences identified by flow cytometry, these data provide additional evidence of an age-associated delay or deficiency in the local cellular immune response to infection in older rhesus macaques.

## Post-acute phase of infection highlights age-stratified systemic immune responses

To expand on our findings from the lungs, we carried out a comparison of systemic immune responses during infection. We observed age-related differences in the changes in circulating frequencies of CD8$^+$ and CD4$^+$ T-cell memory classes post inoculation (Fig S9A and B). Between 3 and 21 dpi, the younger animals exhibited more salient reductions in the frequencies of memory cells in peripheral blood samples, accompanied by a proportional increase in the frequency of naïve cells. This effect was strongest for CD8$^+$ T$_{EEM}$ cells and likely reflects the exodus of effector T cells from the blood to site of infection and/or secondary lymphoid organs. AUC analysis between 1 and 21 dpi confirmed that the reduction in the frequency of this cell type in circulation was greater in the younger animals compared with the older animals ($P$ = 0.0006) (Fig S9A). The principal decrease occurred between 1 and 7 dpi, which aligned with data from the lungs at 7 dpi showing elevated frequencies of T cells, particularly CD8$^+$ T cells, in the tissue of younger animals. Comparison of the frequencies of CXCR3$^+$ cells as a fraction of non-naïve (CD95$^+$) CD8$^+$ and CD4$^+$ T cells over time provided further evidence of more efficient homing of T cells to the site of infection in the younger animals (Fig S9C and D). The augmented frequency of CXCR3 expression was evident on both CD8$^+$ CD95$^+$ and CD4$^+$ CD95$^+$ cells post inoculation, although the effect was stronger for the former (AUC of 1–21 dpi, $P$ = 0.02 and $P$ = 0.04, respectively). In addition, we observed an increase in the frequency of Ki67$^+$ CD8$^+$ CD95$^+$ and CD4$^+$ CD95$^+$ T cells between 7 and 21 dpi; this effect was greater in the younger animals between 14 and 21 dpi (AUC of 1 to 21 dpi, $P$ = 0.03 and $P$ = 0.05, respectively; Fig S9C and D). In summary, these data suggest that the differences in T-cell signatures observed in the lungs stemmed at least in part from the enhanced recruitment of circulating CD8$^+$ T cells to the site of infection in the younger animals.

---

frequency of cells per animal within each cohort are listed to the right of the stacked-bars for each cluster and are calculated using Mann–Whiney $U$ tests. **(D)** Heat map representation of meta-cluster phenotypes. The clusters enriched for the cells of the older (grey), younger (green), and neither (black) are indicated (leftmost column). The major T-cell subset characterizing each cluster is shown by depicting the frequencies of CD4$^+$, CD8$^+$, double negative (DN), and double positive (DP) T cells composing each cluster (middle), as well as a phenotypic comparison of clusters by calculation of the z-score of the median fluorescence intensity of 15 markers across clusters (right). **(E)** Frequency of CD8$^+$ T cells exhibiting a lymph node homing memory phenotype (CD28$^+$ CCR7$^+$) as a percentage of non-naïve CD8$^+$ T cells; and frequencies of CD8$^+$ T cells demonstrating phenotypes associated with tissue-residency (from left to right: CD103$^+$ and CXCR3$^+$ CD69$^+$ CD103$^+$) as a percentage of non-naïve CD8$^+$ T cells are shown. Bars depict median frequencies. $P$-values are calculated using Mann–Whiney $U$ tests. **(F)** Representative multiplex immunohistochemistry (mIHC) of a region of a lung section from an older (left) and a younger (right) rhesus macaque focusing on identified bronchus-associated lymphoid tissue (BALT); α-smooth muscle actin, prosurfactant C, DAPI, HLA-DR, CD3, Ki-67, EpCAM, and CD4 are shown. Some features of the images are denoted such as a large airway (*), germinal center formation (yellow arrow), high endothelial venule (HEV) (white arrowhead), and CD4$^+$ T-cell accumulation (white arrow). Scale bars represent 60 μm. **(G)** Quantification of mIHC images in older and younger animals. Graphs show the number of BALT found in the lung section for each animal (left), the density of Ki-67$^+$ nuclei in all identified BALT (middle), and the area of all identified BALT (right). The blue boxes show the cutoffs for immature BALT (outside shaded area) compared with more advanced BALT (inside shaded area). In panels separated by age-group, grey represents older (O) and green represents younger (Y) rhesus macaques.

Finally, we compared the functional profiles of antigen-specific, systemic T cells at 21 dpi by intracellular cytokine staining of splenocytes stimulated with SARS-CoV-2 spike protein–peptide pools. In support of the above findings, the older animals demonstrated weaker virus-specific pro-inflammatory CD8$^+$ T-cell responses, as evidenced by reduced frequencies of IFNγ- and TNFα-expressing cells compared to those in the younger animals ($P$ = 0.03 for both comparisons; Fig S9E). A similar, but less notable effect was observed for CD4$^+$ IFNγ$^+$ ($P$ = 0.03) and TNFα$^+$ ($P$ = 0.11) T-cell responses (Fig S9F). In conjunction with the previous findings, these data suggest that virus-induced T-cell responses may not only be delayed, but also of inferior quality in the older cohort.

## Age-related differences in the immunomodulatory signaling environment

The previous findings highlighted age-specific differences in inflammation and cellular changes within the lungs, along with kinetic and functional incongruity in T-cell responses. These results prompted us to monitor patterns in the immunomodulatory signaling environment during infection. To this end, levels of serum cytokines were assessed during the acute (1–7 dpi) and the post-acute phases of infection (10–21 dpi). Principal component analysis (PCA) revealed little evidence of sample clustering according to age through 7 dpi (F = 2.5, $P$ = 0.04). However, from 10 dpi onwards, cohort separation became clearer (F = 6.2, $P$ < 0.0001), suggestive of the development of distinct cytokine profiles (Fig 6A). We identified seven cytokines representing the most likely drivers of the age-associated variation observed between 10–21 dpi by applying a loadings cutoff of 0.2 and selecting for high loading scores along PC1: IL-2, IL-10, IL-17A, IL-13, IL-6, and macrophage inflammatory protein (MIP)-1α (Fig 6B). To further examine the conclusions from the PC analysis, we directly compared the concentrations of these cytokines across the sampling time course. In alignment with the PCA data, no dramatic differences in the levels of individual serum cytokines were observed between the age cohorts from 1 to 7 dpi. However, at 10 dpi, concentrations of circulating pro-inflammatory cytokines IL-6, MIP-1α, TNFα, and IL-17A were elevated in the older animals ($P$ = 0.0002, 0.002, 0.06, and 0.05, respectively; Fig 6C). Concurrent spikes in the concentrations of regulatory cytokines IL-10 ($P$ = 0.03) and IL-13 ($P$ = 0.001) were also apparent in the older animals, whereas being absent in the younger. Notably, elevations in the levels of IL-6, IL-10, and IL-13 were sustained in the older animals from 10 dpi through at least 21 dpi (10–21 dpi AUC comparison of old versus young: $P$ = 0.009, 0.02, and 0.03 respectively), suggesting prolonged systemic inflammation.

Because of their role in regulating the activation and resolution phases of the inflammatory response, changes in immune LMs were also assessed at 3, 7, and 21 dpi (Fig 1A). No acute changes in the circulating LM profiles were observed within the first few days after inoculation (3 dpi), although the nonspecific elevation of LMs present in the older animals before infection was preserved (Fig 6D). At 7 dpi, the LM profile of the older animals remained unchanged, but the younger animals began to display a trend of higher LM levels across multiple LM families in agreement with the observed inflection in cytokine behavior on 7 dpi. By 21 dpi, LM levels continued to increase in the younger cohort, whereas

remaining unchanged in the older cohort (Fig 6D). The responsive LMs in the younger animals were primarily localized to the cyclooxygenase family and the 12- and 15-lipoxygenase families (fold change versus 0 dpi and PCA loading) (Fig 6D–F). Increases in similar LM and LM pathway signatures have been associated with an up-regulation of tissue-repair processes. In combination with the findings in the lungs, the differences in LM profiles suggest that by 21 dpi the younger animals had efficiently transitioned away from an acute antiviral response to a pro-resolving or repair-promoting state, as would be expected during convalescence (32, 33, 34).

# Discussion

Increased age is a clear risk factor for severe COVID-19 in humans. Although the mechanism underlying this susceptibility is not fully understood, it is likely driven by impaired and/or dysregulated immune responses to infection in older individuals (3). Investigations into the effects of age on SARS-CoV-2 pathogenesis have been conducted in COVID-19 patients; however, research in humans is limited by sampling constraints and the lack of controlled infection conditions. There is a paucity of data specifically monitoring disease processes in the lungs, during both acute infection and the juncture at which patients recover or proceed to develop more severe disease. In addition, there is a gap in our understanding of the correlates of disease severity in nonhospitalized COVID-19 patients. By using the rhesus macaque model and multi-omics sample analyses, we were able to compare immune dynamics in older and younger animals on a cellular level, with a particular focus on the lungs during two key phases of infection: the acute phase and the post-acute phase.

In our model, similar to previous studies in nonhuman primates (8, 9, 35), clinical disease was mild in both age-groups with older animals demonstrating a slight elevation in disease signs and a longer time to recovery. Despite this limited effect of age on disease outcome, we found several striking differences in the response to SARS-CoV-2 infection through immunological and transcriptional profiling (summarized in Fig 7). During the acute phase of SARS-CoV-2 infection in the older animals, the local innate response was up-regulated in both immune and nonimmune cell populations through 7 dpi. By contrast, the younger animals appeared to better control the innate inflammatory response in the lungs, as evidenced by the more attenuated induction of innate pathways that occurred only in immune cells, and had nearly resolved by 7 dpi. During the post-acute phase of the disease, the two groups diverged further, with older animals exhibiting a prolonged pro-inflammatory circulating response, as well as a delay in the activation and/or differentiation of T cells isolated from the lungs. Concomitant with these changes within the T-cell compartment, the younger animals displayed a pro-resolving circulating milieu of LMs and cytokines at later time points, whereas the older animals did not (Fig 7). Collectively, our findings suggest that increasing age is associated with a prolonged inflammatory state and the delayed or subpar induction of cellular immune responses after SARS-CoV-2 infection.

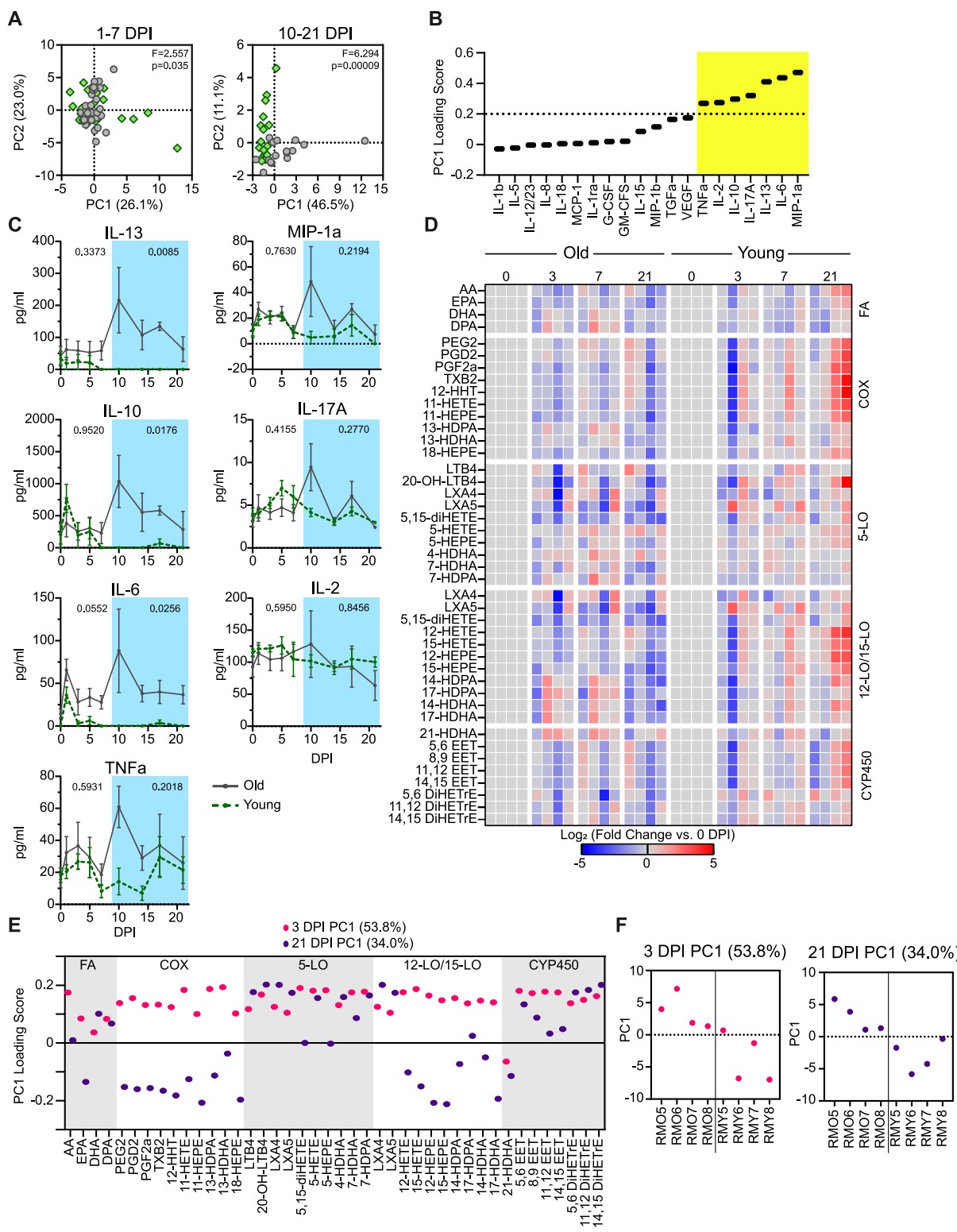

**Figure 6. Circulating cytokines and lipid mediators (LMs) during the acute and post-acute phases of SARS-CoV-2 infection.**
**(A)** Principal component analysis of serum cytokines in the acute phase of the disease (1–7 dpi) and the post-acute phase of disease (10–21 dpi). The first principal component is on the x-axis and the second is on the y-axis. The f-statistic and P-value for clustering across all significant principal components (up to 99% explained variance) is shown from a PERMANOVA test. Green points represent younger and grey represent older animals. **(B)** PC1 loading scores for serum cytokines in the post-acute phase of the disease. The vertical dashed line represents the cutoff for the significant loading values highlighted in the yellow box. **(C)** Serum cytokine levels over the course of infection in the older and younger animals for the cytokines that pass the threshold in panel (B). P-values in the acute phase of the disease (0–7 dpi, outside blue box) and the post-acute phase of the disease (10–21 dpi, inside blue box) represent a comparison of the area under the curve between the older and younger animals

The initiation of the innate immune response is vital to controlling acute viral infections. Studies in humans suggest that younger individuals achieve superior control of SARS-CoV-2 replication and reduced severity of disease through an earlier induction of innate immune responses, especially in the upper respiratory tract (36). In contrast, the rapid initiation of lung innate responses was observed in both age cohorts in our study, with elevated inflammatory signaling present in the older versus younger animals. This finding may explain the self-limiting nature of SARS-CoV-2 infection in the rhesus macaque model and the lack of severe disease, even in aged animals. However, innate responses resolved more slowly in the aged animals compared with the younger macaques. Prolonged inflammation has been linked to the dysregulation of adaptive immune responses, which play a critical role in the resolution of disease (37). Together, these observations suggest that rather than being dependent on the initiation of innate responses, differences in infection dynamics in the rhesus macaque model are driven by the inability to properly dampen the initial innate inflammatory response and a failure to support rapid adaptive responses.

Defective or dysregulated innate and adaptive immune responses may contribute to the worsening COVID-19 outcomes observed with increased age (3). Elevated concentrations of cytokines correlated with severe COVID-19, including IL-6 and IL-10 (38), were present for a longer duration in the older animals than in the younger animals, along with protracted clinical signs. The timing and quality of virus-induced T-cell responses are also predictive of the outcomes of infection for multiple pathogens and are vulnerable to age-related disruption (39). The efficient stimulation of virus-specific IFN-γ–secreting T cells is associated with decreased COVID-19 severity in humans (40). In addition, CD8$^+$ T cells have been found to be among the first cells to enter the lungs after influenza A virus infection and are important in the resolution of disease (41). Indeed, rapid CD8$^+$ T-cell responses combined with efficient dampening of inflammatory responses are linked to recovery from COVID-19 (42, 43). In alignment with this observation (CD8$^+$), effector T-cell frequencies were increased in the lungs of the younger animals at 7 dpi, whereas frequencies of phenotypically similar cells were not elevated in the older animals until 21 dpi. Similarly delayed T-cell responses, measured in the circulating immune pool, have been associated with secondary waves of disease beyond the point of viral clearance in COVID-19 patients, especially when combined with pro-inflammatory signatures (42). These immune dynamics observed in COVID-19 patients are present in our older rhesus macaque cohort. As such, differences between older and younger animals during the post-acute phase of infection may provide insight into contributing cell populations and immune processes that define the continuous scale of severity observed in mild-to-moderate COVID-19 and the prolonged symptoms seen in many COVID-19 patients (42).

A direct relationship is evident between the mild features of immunosenescence present in the older animals at baseline and the immune dynamics after infection indicative of a moderate, but not yet catastrophic, decline in immune responsiveness and homeostatic control. The delay in local cellular immune responses observed in the older animals may additionally be explained by reduced priming capacity with advancing age because of a reduction in the size of the naïve T-cell pool and intrinsic cellular defects, an effect that may have a stronger toll on CD8$^+$ T-cell responses (44). Meanwhile, the sustained local and systemic inflammation in these animals may be explained by the progressive tendency of aging immune systems to shift towards a pro-inflammatory phenotype due to an imbalance of pro- and anti-inflammatory cytokines and decreased efficiency of inflammation-resolving mechanisms (15). In addition, 12- and 15-lipoxygenase pathways have been linked to recovery of lung tissue after viral infection (45, 46), and loss of these LM classes has been linked to higher instances of ICU admissions in COVID-19 patients (13). The observed up-regulation of the LM pathways in the younger, but not older animals in our study, may have contributed to lagging recovery times and potentially to slower repair of lung tissue in the older animals.

Our ability to significantly distinguish some of the age-related immune dynamics discussed above was limited by the group size (n = 8) used in our study, only half of which were monitored up to 21 dpi, and a male-only cohort of younger animals due to a lack of available subadult females. Moreover, given that the study end point did not extend beyond 21 dpi, we could not ascertain if the differences in the immune responses identified between the age groups resolved or became more pronounced over time. Finally, our study did not include animals of exceedingly advanced age. Therefore, and because additional comorbidities associated with more severe COVID-19 in humans and linked with increased age (1) were absent in the animals of our study, we cannot rule out the possibility that significantly worse clinical outcomes would have been observed in severely aged animals with compromised innate immune defenses.

Despite these limitations, the utilization of a large set of high-parameter techniques in one study to generate a time-resolved, multi-site profile of virus-induced innate and adaptive immune responses, has resulted in a comprehensive characterization of SARS-CoV-2 infection in the rhesus macaque model. A thorough understanding of the immune response to SARS-CoV-2 infection in the rhesus macaque model will be essential as we move forward to the next phase of the pandemic. Novel SARS-CoV-2 vaccines and treatments may need to be evaluated without the possibility of performing large phase III efficacy trials because of a lack of cases or most of the susceptible population being vaccinated already. Detailed information from animal models, such as those presented here, will become necessary to enable immunobridging. Importantly, our findings highlight essential differences and similarities

using an unpaired *t* test. Lines represent the mean and error bars represent the SEM at each time point. **(D)** Circulating LM profiles for the older (left) and younger (right) animals over the course of the infection (0, 3, 7, and 21 dpi). Each column represents a single animal, and the rows indicate specific LM. LMs are grouped by the synthesizing lipoxygenase (LO) enzyme noted on the right of the heat map. LM species produced by multiple LO enzymes are listed with each family. Values represent the log$_2$ –fold change compared to day 0 with blue meaning a decrease and red meaning an increase. **(E)** PC1 loading values for the indicated LMs at 3 and 21 dpi. Boxes represent groupings of different LM classes denoted to the right of each group. **(F)** PC1 values for individual animals at 3 and 21 dpi with old on the left of the line and young to the right of the line.

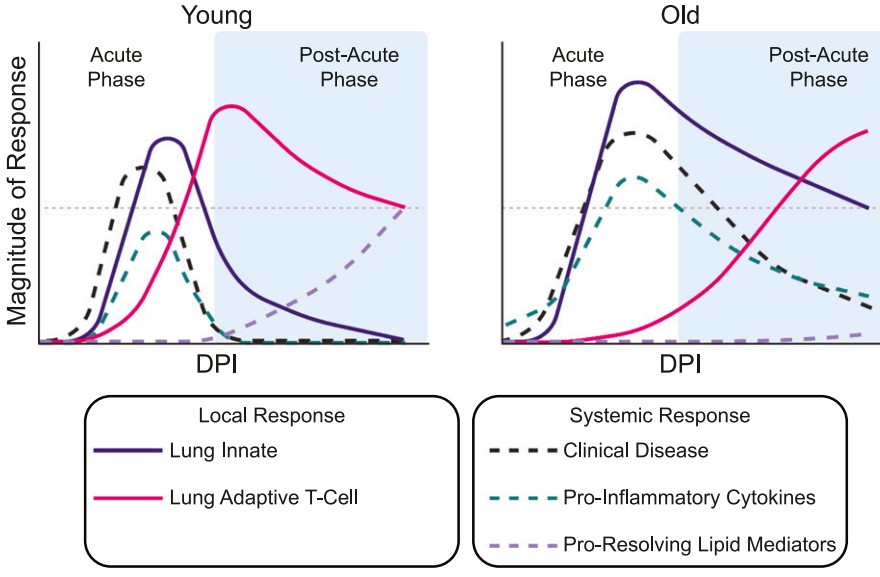

**Figure 7. Summary of the immune dynamics in younger and older rhesus macaques during SARS-CoV-2 infection.**
Graphical representation of the relative magnitude of response over the course of SARS-CoV-2 infection in younger (left) and older (right) rhesus macaques. The white part of each graph indicates the acute phase of the disease and blue is the post-acute phase. The grey horizontal line is for reference between the two graphs at the same point. The different curves represent different local (lung innate and lung adaptive T cell) and systemic (clinical disease, pro-inflammatory cytokines, and pro-resolving lipid mediators) host responses to SARS-CoV-2 infection.

in the immune response to SARS-CoV-2 infection between rhesus macaques and humans that may be affected by age. As such, our study provides new insights into the age-related immune dynamics of SARS-CoV-2 infection and represents a substantial advance in available models of age-associated changes in immunity.

## Materials and Methods

### Study design

To evaluate the effect of age on the pathogenesis of SARS-CoV-2 infection, eight aged (16–23 yr) and eight subadult (3–5 yr) rhesus macaques were inoculated with SARS-CoV-2. Of note, the lifespan of rhesus macaques in captivity is around 35 yr (47) and the comparative age in humans would be 9–15 yr for the younger animals and 48–69 for the older animals (48). All animals were inoculated via a combination of intranasal (0.5 ml per nostril), intratracheal (4 ml), oral (1 ml) and ocular (0.25 ml per eye) routes with a $4 \times 10^5$ $TCID_{50}$/ml ($3 \times 10^8$ genome copies/ml; total dose $2.6 \times 10^6$ $TCID_{50}$ per animal) virus dilution in sterile DMEM. The animals were observed daily and scored for clinical signs using a standardized scoring sheet (35); the same person assessed the animals throughout the study. Clinical exams were performed under anesthesia on 0, 1, 3, 5, 7, 10, 14, 17, and 21 dpi. On examination days, clinical parameters including body weight, body temperature, and respiration rate were recorded, and ventro-dorsal and lateral chest radiographs obtained. BCS was assessed by a clinical veterinarian during each examination and was expressed on a scale of 1–9 (49). A BCS of 4–5 is considered ideal, with scores of 1–3 indicative of an underweight condition and scores of 6–9 indicative of an overweight condition. Of note, BCS is not incorporated in the daily scoring for clinical signs. Blood and swabs (nasal, throat, and rectal) were collected during all clinical exams. In addition, on −7, 3, and 7 dpi, animals

were intubated and bronchoalveolar lavages were performed using 10–20 ml sterile saline. Four animals from each age-group were euthanized at 7 and 21 dpi. After euthanasia on 7 and 21 dpi, necropsies were performed, and tissues were collected.

### Ethics and biosafety statement

All animal experiments were approved by the Institutional Animal Care and Use Committee of Rocky Mountain Laboratories, NIH, and carried out by certified staff in an Association for Assessment and Accreditation of Laboratory Animal Care International accredited facility, according to the institution's guidelines for animal use, following the guidelines and basic principles in the NIH Guide for the Care and Use of Laboratory Animals, the Animal Welfare Act, United States Department of Agriculture, and the United States Public Health Service Policy on Humane Care and Use of Laboratory Animals. Rhesus macaques were housed in adjacent individual primate cages allowing social interactions, in a climate-controlled room with a fixed light–dark cycle (12-h light/12-h dark). Animals were monitored at least twice daily throughout the experiment. Commercial monkey chow, treats, and fruit were provided twice daily by trained personnel. Water was available ad libitum. Environmental enrichment consisted of a variety of human interaction, manipulanda, commercial toys, videos, and music. The Institutional Biosafety Committee (IBC) approved work with infectious SARS-CoV-2 strains under Bio-safety level 3 (BSL3) conditions. Sample inactivation was performed according to IBC-approved standard operating procedures for removal of specimens from high containment.

### Virus and cells

SARS-CoV-2 isolate nCoV-WA1-2020 (MN985325.1) (50) (Vero passage 3) was kindly provided by the CDC and propagated once in VeroE6 cells in DMEM (Sigma-Aldrich) supplemented with 2% FBS (Gibco),

1 mM L-glutamine (Gibco), 50 U/ml penicillin, and 50 μg/ml streptomycin (Gibco). Next generation sequencing using Illumina MiSeq, showed that the virus stock used was 100% identical to the initially deposited GenBank sequence (MN985325.1); SNPs were not present in more than 5% of sequence reads and no contaminants were detected. VeroE6 cells were maintained in DMEM supplemented with 10% fetal calf serum, 1 mM L-glutamine, 50 U/ml penicillin, and 50 μg/ml streptomycin.

## Quantitative PCR

RNA was extracted from swabs and BALF using the QiaAmp Viral RNA kit (QIAGEN) according to the manufacturer's instructions. Tissues (30 mg) were homogenized in RLT buffer and RNA was extracted using the RNeasy kit (QIAGEN) according to the manufacturer's instructions. 5 μL of RNA were used in a one-step real-time RT-PCR assay to detect gRNA and sgRNA (51, 52) using the Rotor-Gene probe kit (QIAGEN) according to instructions of the manufacturer. In each run, standard dilutions of counted RNA standards were run in parallel to calculate copy numbers in the samples.

## Multiplex immunohistochemistry

Rhesus macaque tissues were fixed for a minimum of 7 d in 10% neutral-buffered formalin and embedded in paraffin. Multiplex immunohistochemistry (mIHC) was performed using the IBEX method modification for over-fixed tissue samples (53). The middle right lung lobe of each animal was selected for mIHC analysis. After sectioning, slides were de-waxed according to a standard protocol of two washes of Xylene (Newcomer Supply) for 10 min each, 100% ethanol (Decon Labs Inc.) wash for 10 min, 95% ethanol wash for 10 min, 70% ethanol wash for 5 min, a rinse in water, 10% formalin wash for 15 min, a rinse in water, and stored in TBST solution (1X TBS with 0.05% tween). After de-waxing, antigen retrieval was performed using AR6 buffer (Akoya Biosciences) in a standard microwave with 45 s at 100% power and 15 min at 10% power. Samples were left to cool and rinsed in TBST. A single primary antibody was added at a time. For each primary antibody, the following process was performed. First, the antibody was reacted with the tissue using the microwave method described in the IBEX method (53), then washed in TBST. A mixture of anti-Mouse + anti-Rabbit HRP-conjugated secondary antibodies (Akoya Biosciences) was added for 10 min and then washed in TBST. Finally, the assigned Opal dye (Akoya Biosciences) was added for 3–10 min (see antibody Supplemental Data 2) and then washed in TBST. After Opal dye deposition, antigen retrieval was performed again on samples and the process was repeated using the next primary. At the end of this sequence using a defined set of primary antibodies ("panel"), the slides were mounted with Fluoromount-G (Southern Biotch) and imaged on the Leica Thunder system using a 20× oil objective with an additional autofluorescence channel collected to serve as a fiducial. After imaging of each panel, the coverslips were removed by soaking the slide overnight at room temperature in TBST. To remove fluorescent signal, LiBH$_4$ solution was added to the slides in three washes of 10 min each. Tissue was then stained and imaged sequentially using additional panels stained until all desired markers were evaluated.

DAPI was added to the final panel only. For list of antibodies used, please see the Supplemental Data 2.

After imaging, image files from the different panels were aligned based on autofluorescence using the Imaris Extension software developed for IBEX (https://github.com/niaid/imaris_extensions). Only linear modifications were made to all images as well as Gaussian blurring to reduce noise in channels. All modifications were applied to the entire image and never a single area alone. Quantification of images such as selection of Ki67$^+$ nuclei, segmentation of DAPI signal, identification of cell types, and quantification of BALT structures was performed in Imaris V9.5.0.

## Cytokine and chemokine data collection and normalization

Serum samples for analysis of cytokine/chemokine levels were inactivated by γ-radiation (2 MRad) according to standard operating procedures. Concentrations of granulocyte colony-stimulating factor, granulocyte-macrophage colony-stimulating factor, IFN-γ, IL-1β, IL-1 receptor antagonist, IL-2, IL-4, IL-5, IL-6, IL-8, IL-10, IL-12/23 (p40), IL-13, IL-15, IL-17, monocyte chemoattractant protein-1, MIP-1α, MIP-1β, soluble CD40-ligand (sCD40L), TGF-α, TNF-α, VEGF, and IL-18 were measured on a Bio-Plex 200 instrument (Bio-Rad) using the Non-Human Primate Cytokine MILLIPLEX map 23-plex kit (Millipore) according to the manufacturer's instructions. Care was taken to ensure that all experimental conditions (various dpi and age-groups) were distributed among three plates for the serum samples to avoid introducing batch effects.

Spike-in controls were used to create standard curves for each cytokine to convert the signal readout to cytokine concentrations. For each spike-in control, the ratio of the expected value to the predicted value was calculated. To account for batch effects across plates, the variance of the ratio of observed:expected concentration multiplied by 100 was calculated for the 156.25 pg/ml, 625 pg/ml, and 2,500 pg/ml standards. Cytokines for which the summed variance was greater than 100 were removed from the analysis. Cytokine values were normalized across plates as follows. First, an average of the values of the spike-in controls across plates was calculated. Next, within each plate for each of the standards analyzed, a scaling factor was calculated as the ratio of the standard over the mean across plates. Then an average scaling value was calculated across the factors for each cytokine in each plate. These scaling factors were then applied to the experimental readouts for each cytokine on each plate. For values that could not be determined based on the standard curve (below the limit of detection), the readout value was set to zero.

## LC-MS/MS materials

LC-MS/MS-grade water, methanol, isopropanol, chloroform, ammonium acetate, and acetic acid were purchased through Thermo Fisher Scientific. All LM standards were purchased from Cayman Chemical.

## Sample processing for organic and LM extraction

Sample order was randomized throughout each extraction. For bulk lipid extraction, 50 μl of serum was aliquoted directly into 400 μl of ice-cold methanol and 500 μl of ice-cold chloroform was added.

To induce layering, 400 $\mu$l of water was added. Samples were agitated for 20 min at 4°C and centrifuged at 16,000$g$ for 20 min at 4°C. The organic (bottom) fraction was taken to dryness in a Savant DNA120 SpeedVac concentrator (Thermo Fisher Scientific) and stored at –80°C. For LC-MS/MS sample injection, dried samples were resuspended in 500 $\mu$l of 5 $\mu$g/ml butylated hydroxytoluene in 6:1 isopropanol:methanol.

**LMs sample processing and extraction**

LMs were extracted from macaque serum as previously described (13). Briefly 100 $\mu$l of serum was aliquoted on ice into 400 $\mu$l of ice-cold methanol containing 1 ng each of d8-5-HETE, d5-RvD2, d5-LXA4, d4-LTB4, and d4-PGE2. Macromolecules were precipitated for 30 min at –20°C followed by centrifugation at 10,000$g$ for 10 min. The supernatant was collected in a fresh tube.

Oxy-lipid species were selectively extracted via solid phase extraction columns (Sep-Pak 3 ml, 200 mg, C18; Waters Corporation) as previously described (13). Eluted samples were dried under nitrogen and resuspended in 200 $\mu$l of 1:1 water:methanol. For LC-MS/MS analysis, 30 $\mu$l of each sample was injected.

**LC-MS/MS analysis**

Bulk lipid and LM samples were analyzed using a series of targeted multiple-reaction monitoring methods. All samples were separated using a Sciex ExionLC AC system and analyzed using a Sciex 5500 QTRAP mass spectrometer.

Lipid samples were analyzed using a previously established HILIC method (13, 54). Samples were separated on a Water XBridge Amide column (3.5 $\mu$m, 3 mm × 100 mm) and eluted using a 12-min binary gradient from 100% 5 mM ammonium acetate, 5% water in acetonitrile apparent pH 8.4 to 95% 5 mM ammonium acetate, 50% water in acetonitrile apparent pH 8.0. Target lipids were detected using scheduled multiple-reaction monitoring. Lipid signals were divided into two methods using either negative mode or positive mode and a separate injection was analyzed for each method. Quality control sample injections were performed every 10th injection to ensure instrument stability.

LMs were analyzed as previously described (13). Samples were gradient-eluted from a Waters Atlantis T3 column (100 Å, 3 $\mu$m, 3 × 100 mm) (A: 0.01% acetic acid in water; B: 0.01% acetic acid in methanol). LM species were detected in negative mode and triggered spectra were collected using enhanced-product ion scans and rolling collision energy. A blank and a standard mix were serially injected every 10 injections. The standard mix consisted of each of the following compounds at 10 ng/ml: RvE3, LXA4, LXA5, LXB4, PGE2, PGD2, PGF2a, TxB2, PD1, RvD5, Maresin 1, LTB4, 5,15-DiHETE, 14-HDHA, 18-HEPE, AA, EPA, and DHA. Spectra and comparison to authentic standards were used to confirm signal identity.

All signals were integrated using MultiQuant Software 3.0.3. Bulk lipid data were filtered with a 50% missing value cut-off and a 30% quality control coefficient of variance. For LM datasets, signal quality was judged visually and signals were normalized to internal standards based on the number of hydroxyl groups in the molecule. For bulk lipid class analysis, signals within each class were summed.

**scRNA-seq sample collection**

Lung tissue for single cell sequencing was processed in a manner similar to a method described previously (23). In short, lung samples were taken at the time of necropsy. Cell suspensions were generated by manually dicing the tissue, followed by enzymatic digestion, cell filtration, and ACK lysis to remove red blood cells. BALF processing was performed as follows. Cells were washed, resuspended in DPBS, and strained using a 70-$\mu$m filter (Falcon). Samples were split for use in single cell sequencing and flow cytometry. Once the BALF and lung samples were in suspension, they were prepped according to the 10X genomics protocol for gel bead in emulsion creation. All samples were handled so that approximately 10,000 cells were captured as 10X Genomics GEMs. The 10X Genomics version 3.1 chemistry was used to generate barcoded cDNA and to generate final libraries according to the manufacturer's protocol. After the final libraries were generated, the samples were removed from containment following inactivation of the libraries. Inactivation involved combining 560 $\mu$l of AVL buffer (QIAGEN) with 560 $\mu$l of ethanol and 140 $\mu$l of final library and incubating for a minimum of 10 min. Libraries were extracted from AVL using the QIAGEN AllPrep DNA spin columns (Cat. no. 80204). Samples were then quantified and sequenced using a NextSeq550 using the 10X genomics' suggested cycling conditions.

**scRNA-seq data analysis**

After sequencing, samples were demultiplexed using the cell-Ranger pipeline. Alignments were done against the Rhesus Macaque Genome (Mmul_10, GCA_003339765.3) with the SARS-CoV-2 genome added in (Wuhan-Hu-1, NC_045512.2) to parse out reads to the host versus the virus. Samples were then read into R (v3.6.2) using Seurat (v.3.1.5) (55). BALF and lung samples were processed separately. To account for the differences in sample collection days across the two groups, sample integration was performed to help remove potential batch effects using the IntegrateData function in Seurat. Cells were filtered in a manner similar to the method described previously (23). In short, cells were filtered for high mitochondrial genes and abnormally high or low unique molecular identifier counts. Cells that were likely doublets were relabeled and saved elsewhere. The top 20 principal components of the data were calculated using the 2,000 genes with the highest variance across the dataset. These principal components were used to calculated and UMAP projections and were used to cluster cells. Gene set enrichment analysis was performed using fgsea (56 *Preprint*) and the MSigDB gene sets.

Cell type identification was performed using the method described in reference 23. Reference data used were from human lung single-cell sequencing samples (57). Cell type identification was performed at the individual cell and cluster level. To identify dividing cells, the CellCycleScoring function in Seurat was used to assign the most likely cell cycle phase of each cell. Dividing cell clusters were then identified as having greater than 98% of the cells in the clusters identified as either the S or G2M phase of the cell cycle.

**Cell sample collection and processing for flow cytometry**

PBMCs were isolated from EDTA-whole blood using Histopaque-1077 density-gradient medium (Sigma-Aldrich) and Leucosep tubes

(Greiner Bio-One) according to the manufacturer's instructions. A single cell suspension of lung cells was obtained by disrupting two pieces of tissue per organ (1 cm) using the gentleMACS dissociation system (Miltenyi) followed by enzymatic digestion with 0.5 mg/ml collagenase type XI and 30 mg/ml DNase I, type IV in Roswell Park Memorial Institute 1640 medium at 37°C for 30 min (Sigma-Aldrich). Undigested tissue was removed by pressing the sample through a 70 mM cell strainer and washing with Roswell Park Memorial Institute 1640 medium supplemented with 10% FBS, 1 mM L-glutamine, 50 U/ml penicillin, and 50 μg/ml streptomycin (R10 medium). Erythrocyte removal was performed using RBC Lysis Buffer (Bio-Legend) for 5 min at room temperature. Cells were subsequently washed twice before resuspension in cryopreservation medium (FBS containing 10% DMSO [Sigma-Aldrich]) and transferred to –80°C storage.

## Sample preparation for flow cytometry

Unless otherwise stated, preparation of cell samples for flow cytometric analysis proceeded as follows. Cells were thawed in R10 medium at 37°C, washed in a buffer of 2% FBS/DPBS, and stained with viability dye diluted in DPBS for 20 min at room temperature. Cells were washed before staining for extracellular markers in 2% FBS/DPBS for 30 min at room temperature. Human Fc Block (BD) was added to all surface staining cocktails. After washing, cells were fixed in 2.5% PFA (Biotium) overnight at 4°C. Cells were washed and resuspended in 2% FBS/DPBS before cytometer acquisition. For samples requiring intracellular staining, cells were permeabilized with Cytofix/CytoPerm solution (BD) for 15 min at room temperature. Intracellular staining was performed in 1X Perm/Wash Buffer (BD) for 45 min at room temperature. Finally, cells were washed and resuspended in 2% FBS/DPBS. Samples were acquired on a BD FACSymphony A5 instrument using BD FACSDiva version 8.0.1 software. Analysis of raw data was performed using FlowJo (Treestar, version 10.6.2). For complete list of antibodies, see Supplemental Data 2.

## Immunophenotyping

PBMCs and lung cells were stained using the Zombie Red Fixable Viability Kit (BioLegend), and BALF cells were treated with the LIVE/DEAD Fixable Blue Dead Cell Stain (Thermo Fisher Scientific) before phenotypic staining as described below.

For a depiction of the flow cytometry gating strategies used in this study, see supplemental figures (Figs S10–S14). Briefly, lymphocytes were identified by gating on forward scatter area (FSC-A) versus side scatter area (SSC-A), doublet discrimination was performed using FSC-A versus forward scatter height (FSC-H), and LIVE/DEAD-negative (and CD14⁻ or CD16⁻) cells were selected (Fig S10). Innate cells were identified from total BALF cells by debris exclusion on FSC-A and SSC-A, doublet discrimination (FSC-A versus FSC-H), and LIVE/DEAD-negative cell selection (Fig S12).

## T-cell phenotyping of PBMC and lung cell samples

The T-cell surface staining cocktail contained the following antibodies: CD3 BV650, CD4 BUV395, CD8 PerCP-Cy5.5, CD28 BUV661, CD38 FITC, CD69 AF700, CD25 BUV496, CD154 BV605, HLA-DR APC-Cy7, CD45RA BUV563, CCR7 BV510, CD95 BUV737, CD49d PE-Cy7, CTLA4 BV785, PD-1 APC, CXCR3 BV421, CD14 PE-Dazzle594, and CD16 PE-CF594. Intracellular staining was performed using Ki67 BV711 and FoxP3 PE. T cells were identified by selecting cells negative for CD14, and CD16 myeloid lineage markers and gating on CD3⁺ cells. CD3⁺ T cells were further classified as CD4⁺ (CD4⁺ CD8⁻), CD8⁺ (CD4⁻ CD8⁺), double positive (CD4⁺ CD8ˡᵒ), or double negative (CD4⁻ CD8⁻). CD4⁺ and CD8⁺ T cells were characterized as exhibiting naïve (CD28⁺ CD95ˡᵒ), effector or effector memory (CD28⁻ CD95⁺), or central memory (CD28⁺ CD95ʰⁱ) phenotypes (Figs S10A and S11A and B).

## B-cell phenotyping of PBMC samples

B-cell surface staining was carried out using the following antibodies: CD3 BV650, CD27 BUV805, CD38 FITC, CXCR3 BV421, CD19 PE, CD20 BV510, CD21 BV786, CD24 APC-Cy7, HLA-DR PE-Cy7, CD95 BUV737, CD138 PerCP-Cy5.5, IgD AF647, IgM BUV395, and IgG BUV563. Total B cells were classified as CD3⁻ CD20⁺. Within this gate, class-switched memory, unswitched memory, and naïve B cells were defined as IgD⁻ IgM⁻, IgD⁺ CD27⁺, and IgD⁺ CD27⁻ CD21⁺, respectively, as previously described (58). Plasma cells were defined as CD38⁺ CD138⁺ within the CD3⁻ CD20⁻ population (Fig S10B).

## Phenotyping of BALF cell samples

BALF cell immunophenotyping was performed using a surface marker staining cocktail consisting of CD14 BV421, CCR7 BV510, CD11b BV605, CD80 BV785, CD86 BV711, CD68 AF488, CD163 APC, CD206 BUV737, CD66abce PE, CX3CR1 PerCP-Cy5.5, CD49d PE-Cy7, HLA-DR APC-Cy7, CD35 BUV615, CD63 PE-CF594, and CCR5 BUV496. Neutrophils (CD11b⁺ CD66abce⁺) and macrophages (CD11b⁺ CD66abce⁻ CD68⁺ HLA-DR⁺) were identified as previously described (59, 60). Lymphocytes were isolated by gating on FSC-Aˡᵒ and SSC-Aˡᵒ cells (Fig S12).

## Intracellular cytokine staining of splenocytes

Freshly isolated splenocytes were plated at a concentration of $1.0 \times 10^6$ cells per well and were stimulated at 37°C for 6 h with SARS-CoV-2 spike protein, S1 or S2, peptide pools (GenScript) at a concentration of 2 μg/ml per peptide, or treated with DMSO (unstimulated control). The above cocktails also contained CD28 BUV661 and CD49d antibodies (each at 1 μg/ml), as well as 0.67 μg/ml of GolgiStop and 1 μg/ml of GolgiPlug (BD). Viability staining was performed with the Zombie Red Fixable Viability Kit (BioLegend). Surface markers were stained for with a cocktail of the following antibodies: CD3 BV650, CD4 BUV395, CD8 PerCP-Cy5.5, CD95 BUV737, CD45RA BUV563, CCR7 BV510, CD14 PE-Dazzle594, and CD16 PE-CF594. Staining for intracellular cytokines involved IFNγ AF488, TNFα BV785, Granzyme B APC, IL-2 PE, IL-4 BV421, and IL-10 BV711 (Fig S13A–C). Total responses to antigenic stimulation were determined as follows: for each cytokine, the frequency of positive cells after S1 and S2 stimulation were adjusted for background, by subtraction of frequencies detected in corresponding DMSO-treated control samples, and subsequently summed. Cells stimulated with PMA/ionomycin (eBioscience Cell Stimulation Cocktail) served as a positive control.

## AIM assay

The AIM assay was performed as previously described (61, 62). Cryopreserved single-cell suspensions from lung tissue were thawed and washed in R10 medium before plating at a concentration of $1.0 \times 10^6$ cells per well. Cells were stimulated at 37°C for 18 h with a cocktail of either SARS-CoV-2 spike protein S1 or S2 peptide pools (GenScript) at a concentration of 2 µg/ml per peptide. Unstimulated control wells were established for each sample by treating cells with DMSO. Cells stimulated with PMA/ionomycin (eBioscience Cell Stimulation Cocktail) served as a positive control. Co-stimulation with CD28 BUV661 and CD49d antibodies (each at 1 µg/ml) was carried out for all wells. Viability staining was performed with the Zombie Red Fixable Viability Kit. Surface markers staining included the following antibodies: CD3 BV650, CD4 BUV395, CD8 PerCP-Cy5.5, CD95 BUV737, CD25 BUV496, CD69 AF700, CD137 PE, OX40 BV711, CD154 BV605, PD-L1 BV421, CD14 PE-Dazzle594, and CD16 PE-CF594 (Fig S14). AIM⁺ cells were identified as OX40⁺ CD137⁺. The frequency of positive cells after S1 and S2 stimulation were adjusted for background, by subtraction of frequencies detected in corresponding DMSO-treated control samples, and subsequently summed.

## Serology

Sera were analyzed by SARS-CoV-2 spike protein (S) ELISA as described previously (35). Briefly, maxisorp (Nunc) plates were coated overnight with 100 ng/well S protein diluted in PBS. Sera were serially diluted in duplicate. SARS-CoV-2-specific antibodies were detected using anti-monkey IgG polyclonal antibody HRP-conjugated antibody (KPL), peroxidase-substrate reagent (KPL) and stop reagent (KPL). Optical density was measured at 405 nm. The threshold of positivity was calculated by taking the average of the day 0 values multiplied by 3.

## Data presentation for flow cytometry

For t-SNE and FlowSOM analyses, data from all animals were down-sampled and concatenated. Subsequently, t-SNE analysis (63 Preprint) was performed in the FlowJo (Treestar, version 10.7) environment on compensated parameters (nearest neighbor = 15, minimum distance = 0.5, iterations = 1,000), FlowSOM (version 2.9) (24) analysis was conducted using compensated parameters and fixed generation of 15 meta clusters (set speed = 3) (63 Preprint). Comparison of absolute fluorescence intensity values across individual cells within major T-cell subsets was performed by concatenating data from CD4⁺ or CD8⁺ T cells across older or younger animals and exporting it to ViolinBox (version 5.1.8) (64).

## Statistics and data analysis

Most statistical analyses were performed in GraphPad V9. All statistical tests used are listed in figure legends. All P-values are listed on graphs except in cases where it becomes difficult to read in which case, we used a cutoff of 0.1 for all P-values. Rounded results of statistical tests are also listed in text to add context to the strength of all claims. When comparing directly between two groups, we used

nonparametric tests as estimation of underlying distributions was not possible. When more than two groups were present, a two-way ANOVA was used. Reported P-values are all Benjamin–Hochberg corrected. For sequencing data comparison of gene module scores, a bootstrapping method to estimate P-values was used with 1,000 iterations.

## Data Availability

All data for graphs containing fewer than 16 points per condition have been deposited in Figshare: 10.6084/m9.figshare.16556745. RNA sequencing data have been deposited in the National Center for Biotechnology Information (NCBI) Gene Expression Omnibus and are accessible through Gene Expression Omnibus Series accession number GSE183579.

## Supplementary Information

## Acknowledgements

The authors would like to thank the veterinary and animal care staff of the Rocky Mountain Veterinary Branch for help with animal procedures and animal care; and the Rocky Mountain Laboratories Genomics Unit of the Research Technologies Branch, NIAID, NIH for help with sequencing of samples. This work was supported by the Intramural Research Program of NIAID, NIH.

### Author Contributions

E Speranza: conceptualization, resources, data curation, formal analysis, investigation, visualization, methodology, and writing—original draft, review, and editing.
JN Purushotham: conceptualization, resources, data curation, formal analysis, investigation, visualization, methodology, and writing—original draft, review, and editing.
JR Port: data curation, formal analysis, investigation, methodology, and writing—review and editing.
B Schwarz: resources, data curation, formal analysis, investigation, methodology, and writing—review and editing.
M Flagg: data curation, formal analysis, investigation, visualization, methodology, and writing—review and editing.
BN Williamson: investigation, methodology, and writing—review and editing.
F Feldmann: investigation, methodology, and writing—review and editing.
M Singh: investigation, methodology, and writing—review and editing.
L Perez-Perez: investigation, methodology, and writing—review and editing.
GL Sturdevant: investigation, methodology, and writing—review and editing.
LM Roberts: investigation, methodology, and writing—review and editing.

A Carmody: investigation, methodology, and writing—review and editing.

JE Schulz: investigation, methodology, and writing—review and editing.

N van Doremalen: investigation, methodology, and writing—review and editing.

A Okumura: investigation, methodology, and writing—review and editing.

J Lovaglio: investigation, methodology, and writing—review and editing.

PW Hanley: investigation, methodology, and writing—review and editing.

C Shaia: formal analysis, investigation, methodology, and writing—review and editing.

RN Germain: resources, investigation, methodology, and writing—review and editing.

SM Best: resources, methodology, and writing—review and editing.

VJ Munster: resources, investigation, methodology, and writing—review and editing.

CM Bosio: resources, methodology, and writing—review and editing.

E de Wit: conceptualization, resources, formal analysis, supervision, validation, investigation, methodology, and writing—original draft, review, and editing.

## Conflict of Interest Statement

The authors declare that they have no conflict of interest.

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
