## [Reviewer comments · Life Science Alliance]

Life Science Alliance

Age-related differences in immune dynamics during SARS-CoV-2 infection in rhesus macaques

Emmie de Wit, Emily Speranza, Jyothi Purushotham, Julia Port, Benjamin Schwarz, Meaghan Flagg, Brandi Williamson, Friederike Feldmann, Manmeet Singh, Lizzette Perez-Perez, Gail Sturdevant, Lydia Roberts, Aaron Carmody, Jonathan Schulz, Neeltje van Doremalen, Atsushi Okumura, Jamie Lovaglio, Patrick Hanley, Carl Shaia, Ronald Germain, Sonja Best, Vincent Munster, and Catharine Bosio

DOI: <https://doi.org/10.26508/lsa.202101314>

Corresponding author(s): *Emmie de Wit, National Institute of Allergy and Infectious Diseases*

Review Timeline:	Submission Date:	2021-11-23
	Editorial Decision:	2021-11-24
	Revision Received:	2021-12-17
	Editorial Decision:	2021-12-22
	Revision Received:	2022-01-04
	Accepted:	2022-01-05

Transaction Report:

Please note that the manuscript was previously reviewed at another journal and the reports were taken into account in the decision-making process at *Life Science Alliance*.

Reviewer #1 Review

Comments to the Author (Required):

In this study Rhesus macaques of distinct ages (8 young animals ; age 3-4 years/ 8 old : age 18 years) were infected with SARS-CoV-2 . Immunological parameters were analyzed in serum and PBMC (0,1,3,5,7, 14 and 21) bronchoalveolar lavage (=7 3 ,7 Days) and lung (day 7)). Spleen cells were also analyzed at day 21. In addition to test cytokines and lipid in sera, the authors analyzed immune cells (monocytes, T and B cells) activation/proliferative state in PBMC and BALF. They also performed a single cell RNA analysis of immune cells present in the BALF and Lung. They also test Spike-specific T cells in spleen cells at day 21. The study might be of some interest for the investigation of the possible causes of the high incidence of Severe COVID-19 i in older people.

1) The fact that the pathology induced by SARS-CoV-2 infection in these animals is identical in old and young macaques triggers reservations about the robustness of the model and the importance of the observations derived.

2) The authors claim that their data show that age may delay and impair induction of antiviral cellular immune response and a delay return to immune homeostasis

I have reservation about this claim since there is no longitudinal analysis of any virus-specific immune response in all the paper.

SARS-specific antibodies or T cells have not been analyzed at all and the only analysis of SARS-CoV-2 specific adaptive immunity is a test of Spike-specific T cells performed at day 21. As such the study shows a large quantity of unspecific parameters of activation/proliferation detected through single cell Gene expression or with phenotypic analysis of cells. Such activation are interpreted by the authors as possible signs of induction of virus-specific immunity. This is possible, but this work should demonstrated that such unspecific parameters are linked with virus-specific immune response.

3) The longitudinal data of single cells analysis in BALF (Figure 3) might be of some interest since show a persistence of inflammatory signature in older macaques, but again the complete absence of any analysis of virus-specific adaptive immunity in the animals (not necessary in the BALF) do not allow to really understand whether such persistence inflammation was due to different level or quality of T and antibody response at earlier time points. It is not clear why the authors do not perform a longitudinal analysis of humoral and cellular immunity at least in the serum/ PBMC of the infected animals .

Reviewer #2 Review

Comments to the Author (Required):

In the manuscript entitled "age-related differences in immune dynamics during SARS-CoV-2 infection in rhesus macaques", the authors utilized two cohorts of macaques with different ages to investigate the effect of age on immune responses to SARS-CoV-2 infection. By multi-omics analysis, the authors concluded that age may delay or impair the induction of anti-viral cellular immune responses and delay efficient return to immune homeostasis. However, the novelty of this manuscript was a little poor.

1. In the manuscript, although the authors quantified SARS-CoV-2 RNA level, they did not assess virus-specific B and T cells. It is more informative to add those data to make the conclusion.

2. In Figure 1, how about the relationship between obesity of animals and immune response to SARS-CoV-2? Additionally, why did the cohort with old macaques include male and female while the young cohort only includes male? Could this affect the results?

3. In Figure 3, the authors claimed that the frequency of BALF T-cells was elevated in the younger animals." How about cell number? The elevated frequency of T cells may be due to the significant change of ciliated and pneumocytes as shown in Fig. 3c. The absolute cell number should be shown.

4. In Lines 172-174: "gene modules 3 and 5 refer to type-I-interferon responsive". One recent study on rhesus macaques found that genes associated with Notch signaling and type-I-interferon were more upregulated in the lung of younger animals 14 days after SARS-CoV-2 infection. You can compare and discuss your results with this paper.

Rosa, B.A., Ahmed, M., Singh, D.K. et al. IFN signaling and neutrophil degranulation transcriptional signatures are induced during SARS-CoV-2 infection. *Commun Biol* 4, 290 (2021). <https://doi.org/10.1038/s42003-021-01829-4>.

Another recent study reported that age-related effects were more pronounced in baboons than macaques. Comparison and discussion should be carried out. (Singh, D.K., Singh, B., Ganatra, S.R. et al. Responses to acute infection with SARS-CoV-2 in the lungs of rhesus macaques, baboons and marmosets. *Nat Microbiol* 6, 73-86 (2021). <https://doi.org/10.1038/s41564-020-00841-4>)

Reviewer #3 Review

Comments to the Author (Required):

In this manuscript Speranza and colleagues investigated the relationship between age and host immune responses to SARS-CoV-2 infection in rhesus macaques (RMs). The authors used two cohorts of eight older and eight younger RMs, infected them with SARS-CoV-2 and measured various parameters associated with viral and immune parameters in the local and systemic compartments throughout the course of disease. These include viral loads, clinical parameters, thoracic radiographs, single-cell transcriptomics, multiparameter flow cytometry, multiplex immunohistochemistry, plasma cytokines, and lipidomics. The authors performed the sample collections and analyses at the acute phase, post-acute phase, and the transition point between the two phases. The authors used cutting-edge technologies to dwell into differences between groups at molecular and single cell level in the lung. The analyses are comprehensive, carefully done and the results are presented well. Although, the study showed limited effects of age on disease outcome, there are several striking differences in the response to SARS-CoV-2 infection. These include clinical condition, kinetics of lipid metabolism, cytokine regulation, and adaptive immune responses that have the potential to impact control of infection. Overall, the study showed some interesting longitudinal changes in cell subsets and their phenotypes that occur in the lung and revealed that older animals show a slight elevation in clinical score and take longer to recover compared to young animals. I have the following three minor comments:

1) The authors presented age-related differences in cell frequencies at 7 dpi in BALF, with the older animals displaying higher frequencies of epithelial populations, specifically ciliated cells (CC) and pneumocytes (PC). Some of the recent studies have observed viral RNA in these cell types. Do the authors have any data on the infection status of these cells if so, it would be good to include these data or comment.

2) It would be informative to include a statement on how the age of a monkey relates to human age.

3) Fig S8a, the axis labels seems to be switched for the flow plot showing CD3 Vs Ki67.

November 24, 2021

Re: Life Science Alliance manuscript #LSA-2021-01314-T

Dr. Emmie de Wit
National Institute of Allergy and Infectious Diseases
Hamilton, MT

Dear Dr. de Wit,

Thank you for submitting your manuscript entitled "Age-related differences in immune dynamics during SARS-CoV-2 infection in rhesus macaques" to Life Science Alliance. We invite you to submit a revised manuscript addressing the Reviewer comments.

Thank you for this interesting contribution to Life Science Alliance. We are looking forward to receiving your revised manuscript.

Sincerely,

Eric Sawey, PhD
Executive Editor
Life Science Alliance
<http://www.lsa-journal.org>

B. MANUSCRIPT ORGANIZATION AND FORMATTING:

Dear Editor,

We thank the reviewers for their helpful suggestions. Below and in the revised manuscript, we have addressed the issues raised. This has resulted in several additional analyses and 2 new display items. The data from our manuscript have been deposited in Figshare ([10.6084/m9.figshare.16556745](https://www.figshare.com/figure/10.6084/m9.figshare.16556745)), in Supplemental File 1, and in NCBI's Gene Expression Omnibus (GEO Series accession number GSE183579). We hope that you agree that we have sufficiently responded to the reviewers' concerns and our manuscript is now suitable for publication in LSA.

Best,
Emmie de Wit, PhD

Reviewer #1:

In this study Rhesus macaques of distinct ages (8 young animals ; age 3-4 years/ 8 old : age 18 years) were infected with SARS-CoV-2 . Immunological parameters were analyzed in serum and PBMC (0,1,3,5,7, 14 and 21) bronchoalveolar lavage (=7 3 ,7 Days) and lung (day 7)). Spleen cells were also analyzed at day 21.

In addition to test cytokines and lipid in sera, the authors analyzed immune cells (monocytes, T and B cells) activation/proliferative state in PBMC and BALF. They also performed a single cell RNA analysis of immune cells present in the BALF and Lung. They also test Spike-specific T cells in spleen cells at day 21. The study might be of some interest for the investigation of the possible causes of the high incidence of Severe COVID-19 i in older people.

1) The fact that the pathology induced by SARS-CoV-2 infection in these animals is identical in old and young macaques triggers reservations about the robustness of the model and the importance of the observations derived.

While the reviewer is correct that the disease signs observed in young and older rhesus macaques are similar, it is worth noting that we did observe a moderate increase in clinical scores in the older animals that lasted for longer and moderate increases in the pulmonary infiltrates observed in x-rays (Fig. 2a-b; Table S1). This is in line with other studies in aged rhesus macaques (as discussed in lines 65-71 and 395-397) that have not shown large differences in clinical presentation, so the model is robust in that sense. Although the model is not reflective of the severe disease observed with increased frequency in older patients, the rhesus macaque is the only relevant SARS-CoV-2 infection model that can be used for the study presented here. Other models lack reagents (e.g. hamsters, ferrets), or do not represent a natural infection (mice; K18-hACE2 or mouse-adapted SARS-CoV-2). Our study goes beyond the clinical presentation and has examined the infection dynamics and immune response on a level not previously presented. Important hallmarks of an aging immune system like those observed in humans are observed in our model (Fig. 1). Our principal finding is that despite similar clinical presentation across animals, an age-related divergence in the immune response became apparent at the point of resolution of clinical signs and grew more pronounced in the post-acute phase of infection. The time at which the immune responses began to diverge aligned with the timing of clinical resolution of disease in the younger animals and sustained disease in the older animals. Additionally, although the study did not result in a severe disease model as we had hoped,

we are providing the most in-depth characterization of SARS-CoV-2 infection and immune response dynamics in a SARS-CoV-2 infection model to date. Therefore, we respectfully disagree with the reviewer on the (lack of) importance of our manuscript.

2) The authors claim that their data show that age may delay and impair induction of antiviral cellular immune response and a delay return to immune homeostasis

I have reservation about this claim since there is no longitudinal analysis of any virus-specific immune response in all the paper. SARS-specific antibodies or T cells have not been analyzed at all and the only analysis of SARS-CoV-2 specific adaptive immunity is a test of Spike-specific T cells performed at day 21. As such the study shows a large quantity of unspecific parameters of activation/proliferation detected through single cell Gene expression or with phenotypic analysis of cells. Such activation are interpreted by the authors as possible signs of induction of virus-specific immunity. This is possible, but this work should demonstrated that such unspecific parameters are linked with virus-specific immune response.

The reviewer is correct that we have not shown virus-specific T-cell responses in circulation. One reason for this is the limitation imposed by animal welfare regulations on the amount of blood that can be drawn from animals in repeated blood draws (max. 15% of total blood volume in two weeks at our institute). Since blood was also collected for virological and serological analysis, a limited volume was available for isolation of PBMC. Therefore, we had to make choices in the analyses we could perform on these samples, and we decided to perform the assays shown in Fig. S9. Additional systemic metrics of the immune response, such as cytokine and lipidomics analyses, are provided to address differences in the systemic response to SARS-CoV-2 infection in the two cohorts.

Moreover, we think showing circulating levels of specific T-cells is not as informative as showing the specific T-cells in the lungs at the time when immune responses in the animals diverged (i.e. 7 dpi). Therefore, we have now included data from an Activation Induced Marker (AIM) assay performed on lung tissue collected on 7 dpi (Fig. S8 and lines 257-262). The AIM assay was developed to detect antigen-specific T cells with high sensitivity and was validated for use in rhesus macaques (Reiss et al., Plos ONE 2017). Using this assay, we detect activated SARS-CoV-2-specific T-cells in the lungs of the young and older rhesus macaques; higher frequencies of activated T-cells are detected in the lungs of younger animals and this difference is significant for CD8 T-cells. These differences in virus-specific T-cells are still detectable in the spleen on 21 dpi. Thus, these additional data again support our conclusion that despite similar disease outcomes, age may delay or impair the induction of anti-viral cellular immune responses and delay efficient return to immune homeostasis.

3) The longitudinal data of single cells analysis in BALF (Figure 3) might be of some interest since show a persistence of inflammatory signature in older macaques, but again the complete absence of any analysis of virus-specific adaptive immunity in the animals (not necessary in the BALF) do not allow to really understand whether such persistence inflammation was due to different level or quality of T and antibody response at earlier time points. It is not clear why the authors do not perform a longitudinal analysis of humoral and cellular immunity at least in the serum/ PBMC of the infected animals .

As explained above, we think local responses are more important than systemic responses, especially since the animals develop very little systemic disease. We have now included virus-specific T-cell data in lungs on 7 dpi (Fig. S8 and lines 257-262). As an additional measure of virus-specific immunity, we have now also included serology, showing spike-specific IgG profiles over time. Even though the older animals were slower to recover and had sustained inflammation, spike-specific IgG was detected in serum earlier in older than younger animals (Fig. S2 and lines 142-145). This again highlights the importance of the local, rather than systemic, immune response.

Reviewer #2:

In the manuscript entitled "age-related differences in immune dynamics during SARS-CoV-2 infection in rhesus macaques", the authors utilized two cohorts of macaques with different ages to investigate the effect of age on immune responses to SARS-CoV-2 infection. By multi-omics analysis, the authors concluded that age may delay or impair the induction of anti-viral cellular immune responses and delay efficient return to immune homeostasis. However, the novelty of this manuscript was a little poor.

1. In the manuscript, although the authors quantified SARS-CoV-2 RNA level, they did not assess virus-specific B and T cells. It is more informative to add those data to make the conclusion.

As indicated in our responses to reviewer #1, we have added the AIM assay to assess the virus-specific T cell responses in the lungs and anti-spike ELISA data to address the humoral response to SARS-CoV-2 infection (Fig. S2 and Fig. S8 and lines 257-262 and lines 142-145).

2. In Figure 1, how about the relationship between obesity of animals and immune response to SARS-CoV-2? Additionally, why did the cohort with old macaques include male and female while the young cohort only includes male? Could this affect the results?

The reviewer is correct that the two cohorts of rhesus macaques are not a perfect match. This is the result of availability rather than by choice or design. However, the older rhesus macaques are considered overweight rather than obese by veterinary standards and we cannot detect classical markers of true obesity in macaques such as sphingolipids in our older cohort (lines 99-100). The lack of female animals in our younger cohort is due to the increased demand for rhesus macaques during the pandemic; as a result, female animals are maintained for breeding programs, rather than enrolled in experiments. Though we cannot deconvolve the effects of being overweight or sex, we have addressed these as potential drawbacks in the discussion (lines 456-460).

3. In Figure 3, the authors claimed that the frequency of BALF T-cells was elevated in the younger animals." How about cell number? The elevated frequency of T cells may be due to the significant change of ciliated and pneumocytes as shown in Fig. 3c. The absolute cell number should be shown.

With single cell RNA sequencing, the cell numbers are very low in comparison to what can be achieved through flow cytometry. For this reason, it is not correct to report cell counts as these can be heavily biased by number of total cells identified in the samples. Because of this issue, all the results in relation to cell frequency changes for the sequencing are further validated through flow cytometry (Fig 3b and Figure S4b, Fig 3c and Fig 4d). We agree with the reviewer that changes in frequencies could be due to one population shifting (as stated in lines 158-159).

4. In Lines 172-174: "gene modules 3 and 5 refer to type-I-interferon responsive". One recent study on rhesus macaques found that genes associated with Notch signaling and type-I-interferon were more upregulated in the lung of younger animals 14 days after SARS-CoV-2 infection. You can compare and discuss your results with this paper.

Rosa, B.A., Ahmed, M., Singh, D.K. et al. IFN signaling and neutrophil degranulation transcriptional signatures are induced during SARS-CoV-2 infection. *Commun Biol* 4, 290 (2021). <https://doi.org/10.1038/s42003-021-01829-4>.

Another recent study reported that age-related effects were more pronounced in baboons than macaques. Comparison and discussion should be carried out. (Singh, D.K., Singh, B., Ganatra, S.R. et al.

Responses to acute infection with SARS-CoV-2 in the lungs of rhesus macaques, baboons and marmosets. *Nat Microbiol* 6, 73-86 (2021). <https://doi.org/10.1038/s41564-020-00841-4>

The reviewer is correct that some data comparing young and older rhesus macaques are already available, and these (several additional manuscripts besides the ones mentioned by the reviewer) are referenced in the manuscript (lines 65-71 and lines 395-397). However, these studies suffer from limited animal numbers, a lack of in-depth analyses and a lack of analysis throughout infection. The interferon response we note is a general inflammatory response and is discovered from an unbiased clustering of the genes. Although this was indeed described previously, our analyses were performed on SCSeq data rather than bulk sequencing as in Rosa et al. Moreover, the comparison of older versus young animals in that study is hampered by the fact that the group sizes are different (3 young versus 5 old) and, more importantly, that the animals were not euthanized on the same day post inoculation (14 or 15 dpi for old animals versus 17 dpi for young animals). Besides our SCSeq data, our analyses are further corroborated with flow cytometry, cytokine analysis and lipidomics to obtain a clear picture of the immune dynamics in younger vs older rhesus macaques. We have now included a more specific reference to these previous data in the revised manuscript (lines 183-185).

Reviewer #3:

In this manuscript Speranza and colleagues investigated the relationship between age and host immune responses to SARS-CoV-2 infection in rhesus macaques (RMs). The authors used two cohorts of eight older and eight younger RMs, infected them with SARS-CoV-2 and measured various parameters associated with viral and immune parameters in the local and systemic compartments throughout the course of disease. These include viral loads, clinical parameters, thoracic radiographs, single-cell transcriptomics, multiparameter flow cytometry, multiplex immunohistochemistry, plasma cytokines, and lipidomics. The authors performed the sample collections and analyses at the acute phase, post-acute phase, and the transition point between the two phases. The authors used cutting-edge technologies to dwell into differences between groups at molecular and single cell level in the lung. The analyses are comprehensive, carefully done and the results are presented well. Although, the study showed limited effects of age on disease outcome, there are several striking differences in the response to SARS-CoV-2 infection. These include clinical condition, kinetics of lipid metabolism, cytokine regulation, and adaptive immune responses that have the potential to impact control of infection. Overall, the study showed some interesting longitudinal changes in cell subsets and their phenotypes that occur in the lung and revealed that older animals show a slight elevation in clinical score and take longer to recover compared to young animals. I have the following three minor comments:

1) The authors presented age-related differences in cell frequencies at 7 dpi in BALF, with the older animals displaying higher frequencies of epithelial populations, specifically ciliated cells (CC) and pneumocytes (PC). Some of the recent studies have observed viral RNA in these cell types. Do the authors have any data on the infection status of these cells if so, it would be good to include these data or comment.

The reviewer is correct that it would be interesting to know which specific cell types are infected in the BALF. We have used such analyses before to determine that virus replication is mostly limited to pneumocytes rather than macrophages in the lung of African Green Monkeys (Speranza et al., *Science Translational Medicine*, 2021) and would have liked to include that analysis here. Unfortunately, there were not enough reads in the BALF mapping to the viral genome to address which cell types were supportive of virus replication. This is now explained in the revised manuscripts (line 152-153). At the timepoints we assessed the lungs (7dpi and 21 dpi), there was no longer enough viral material (RNA or

protein) to quantify (in single-cell seq or IHC). Thus, in this study, we cannot address directly if there is replication in ciliated cells.

2) It would be informative to include a statement on how the age of a monkey relates to human age. **We have added human equivalent age ranges to the Methods section (lines 483-484) as requested.**

3) Fig S8a, the axis labels seems to be switched for the flow plot showing CD3 Vs Ki67. **This has been corrected. It has also been moved to Fig. S10a.**

December 22, 2021

RE: Life Science Alliance Manuscript #LSA-2021-01314-TR

Dr. Emmie de Wit
National Institute of Allergy and Infectious Diseases
903 S 4th Street
Hamilton, MT 59840

Dear Dr. de Wit,

Thank you for submitting your revised manuscript entitled "Age-related differences in immune dynamics during SARS-CoV-2 infection in rhesus macaques". We would be happy to publish your paper in Life Science Alliance pending final revisions necessary to meet our formatting guidelines.

- please use capital letters when introducing panels in figures, their legends and callouts in the manuscript text
- Tables can be included at the bottom of the main manuscript file or be sent as separate files
- please add callouts for Figures 2C, D; S11A-B; S13A-C to your main manuscript text

A. FINAL FILES:

B. MANUSCRIPT ORGANIZATION AND FORMATTING:

****It is Life Science Alliance policy that if requested, original data images must be made available to the editors. Failure to provide original images upon request will result in unavoidable delays in publication. Please ensure that you have access to all original**

data images prior to final submission.**

The license to publish form must be signed before your manuscript can be sent to production. A link to the electronic license to publish form will be sent to the corresponding author only. Please take a moment to check your funder requirements.

Sincerely,

January 5, 2022

RE: Life Science Alliance Manuscript #LSA-2021-01314-TRR

Dr. Emmie de Wit
National Institute of Allergy and Infectious Diseases
903 S 4th Street
Hamilton, MT 59840

Dear Dr. de Wit,

Thank you for submitting your Research Article entitled "Age-related differences in immune dynamics during SARS-CoV-2 infection in rhesus macaques". It is a pleasure to let you know that your manuscript is now accepted for publication in Life Science Alliance. Congratulations on this interesting work.

DISTRIBUTION OF MATERIALS:

Again, congratulations on a very nice paper. I hope you found the review process to be constructive and are pleased with how the manuscript was handled editorially. We look forward to future exciting submissions from your lab.

Sincerely,
